# A Middle Pleistocene Denisovan molar from the Annamite Chain of northern Laos

Fabrice Demeter [1,2,33✉], Clément Zanolli [3,33✉], Kira E. Westaway [4], Renaud Joannes-Boyau [5,6], Philippe Duringer[7], Mike W. Morley[8], Frido Welker[9], Patrick L. Rüther [10], Matthew M. Skinner [11,12], Hugh McColl [1], Charleen Gaunitz[1], Lasse Vinner[1], Tyler E. Dunn[13], Jesper V. Olsen [10], Martin Sikora[1], Jean-Luc Ponche[14], Eric Suzzoni[15], Sébastien Frangeul[15], Quentin Boesch[7], Pierre-Olivier Antoine[16], Lei Pan[17,18], Song Xing[17,19], Jian-Xin Zhao [20], Richard M. Bailey [21], Souliphane Boualaphane[22], Phonephanh Sichanthongtip[22], Daovee Sihanam[22], Elise Patole-Edoumba[23], Françoise Aubaile[2], Françoise Crozier[24], Nicolas Bourgon [12], Alexandra Zachwieja[25], Thonglith Luangkhoth[22], Viengkeo Souksavatdy[22], Thongsa Sayavongkhamdy[22,34], Enrico Cappellini [9], Anne-Marie Bacon[26], Jean-Jacques Hublin [12,27], Eske Willerslev[1,28,29,30] & Laura Shackelford [31,32,33✉]

The Pleistocene presence of the genus *Homo* in continental Southeast Asia is primarily evidenced by a sparse stone tool record and rare human remains. Here we report a Middle Pleistocene hominin specimen from Laos, with the discovery of a molar from the Tam Ngu Hao 2 (Cobra Cave) limestone cave in the Annamite Mountains. The age of the fossil-bearing breccia ranges between 164–131 kyr, based on the Bayesian modelling of luminescence dating of the sedimentary matrix from which it was recovered, U-series dating of an overlying flowstone, and U-series–ESR dating of associated faunal teeth. Analyses of the internal structure of the molar in tandem with palaeoproteomic analyses of the enamel indicate that the tooth derives from a young, likely female, *Homo* individual. The close morphological affinities with the Xiahe specimen from China indicate that they belong to the same taxon and that Tam Ngu Hao 2 most likely represents a Denisovan.

---

A full list of author affiliations appears at the end of the paper.

 1

From the Early to Late Pleistocene, the presence of *Homo erectus* is well documented in Asia, notably in China and Indonesia[1–3]. However, the taxonomic attribution of most Asian late Middle Pleistocene *Homo* specimens remains a matter of contention[4–7]. The recent description and analysis of the Harbin cranium from China has reignited this debate by suggesting its attribution to a new species named *Homo longi*[8], but this new taxonomic attribution remains highly debated. In fact, the Harbin cranium shows close morphological similarities with other late Middle to early Late Pleistocene Asian *Homo* specimens from Dali, Xujiayao, Xuchang and Hualongdong, whose taxonomy remains unclear[4,9,10]. These fossils are considered to belong to a different taxon than *H. erectus* and are often grouped under the generic label 'archaic humans'[9,10]. Due to the combination of features they exhibit, including Neanderthal-like traits, it has been suggested that they belong to an Asian sister taxon of Neanderthals, the Denisovans, even if this attribution remains under debate[5,11,12]. The small number of fossils currently securely attributed to the Denisovans (Denisova 2, a lower left molar; Denisova 3, a distal manual phalanx; Denisova 4, an upper left M3; Denisova 8, an upper molar; and the Xiahe mandible)[13–16] prohibits a clear morphological picture of their overall morphology. The geographic distribution of the Denisovans also remains debated. Modern Papuans, Aboriginal Australians, Oceanic/Melanesian, Philippine Ayta groups and, to a much lesser extent, mainland Southeast Asian populations, retain a Denisovan genetic legacy[14,17–19]. Combined paleoproteomic and morphometric analyses recently suggested that the Middle Pleistocene Xiahe mandible from Baishiya Karst Cave belonged to a Denisovan, extending the known range of this group onto the Tibetan Plateau[15]. However, there is still no fossil evidence explaining the Denisovans genetic imprint on modern southeast Asian populations and—due to the paucity of the Middle Pleistocene fossil record—it is still unknown whether one or more human lineages (co)existed in continental southern Asia. We present here to the best of our knowledge the first unambiguous Middle Pleistocene *Homo* specimen from mainland southeast Asia and discuss its taxonomic attribution and implications for human evolution in the region.

In December 2018, a hominin permanent lower molar was recovered from a breccia block at Tam Ngu Hao 2 (Cobra Cave), Huà Pan province, Laos (20°12′41.5″N, 103°24′32.2″E, altitude 1,116 m; Fig. 1, Supplementary Fig. 1). The tower karst in which the cave was formed is positioned on the south-eastern side of P'ou Loi Mountain with an entrance located 34 m above the alluvial plain (Fig. 1a, Supplementary Fig. 1). The site was discovered during a survey of the area around Tam Pà Ling, where early *Homo sapiens* fossils have previously been recovered[20–22]. The tooth (TNH2-1) is a mandibular left permanent molar crown germ (Fig. 2a–f; Supplementary Fig. 2), and the absence of occlusal and interproximal wear combined with the incipient root formation suggests that the tooth was unerupted at the time of the individual's death. The morphology of the tooth is compatible with an attribution to either a first or a second lower molar (Methods-Detailed morphological analysis of the tooth). In either case, considering the early maturational stage of the root, this tooth belonged to a juvenile individual corresponding to an age ranging from 3.5 to 8.5 years following modern developmental standards[23].

In this work to best document THN2-1, morphological description and comparative analyses are performed. We also develop a specific sampling protocol that allows us to sample for palaeoproteomic and future isotopic analyses while preserving the whole occlusal surface morphology of the crown. Sampling for these destructive analyses takes place after microCT analysis of the entire tooth, ensuring full morphological data were saved. No additional sampling for ancient DNA analyses is performed at this stage given the old age of the specimen and the tropical conditions under which the sediment and fossils were deposited. The invasive sampling strategy to collect dental tissues for molecular analyses only focuses on the distal part of the inferior aspect of the crown, keeping the mesial portion of the crown intact.

## Results

**Context and dating**. The geological setting, stratigraphy and micromorphology of the sediment sequence were analysed to obtain a comprehensive, multi-scalar assessment of the depositional context and taphonomic history of the fossils recovered from the cave (Supplementary Information, Geology). The partially eroded sediments that infill the studied entrance passage comprise a lower and an upper facies representing two phases of sediment accumulation separated by an erosional surface and an unknown period of time (Fig. 1b). The lower facies (Lithological Unit 1, LU1) is weakly cemented and forms an arenitic silty clay deposit that is devoid of fossils (Fig. 1e). The upper, fossiliferous facies (Lithological Unit 2, LU2) is well cemented and coarse grained, containing intrakarstic angular limestone clasts and extrakarstic rounded pebbles, forming a very hard breccia/conglomerate layer from which skeletal elements—and in particular, teeth—were recovered in high frequencies (Fig. 1d). The change in lithology between the two facies most likely reflects a reconfiguration of the karstic hydrological system as would be associated with a major flood, eroding space in LU1 onto which the sediments of LU2 were unconformably overlain. The sediments of LU2 are laterally contiguous and densely packed throughout the exposure excavated for this study, precluding major reworking of material and confirming the stratigraphic context of the fossils contained within, including the hominin tooth (see detailed observations described in Methods). The upper facies (LU2) is draped with two carbonate flowstones, indicating a final change in hydrology and the passage of surface water out of the cave and the precipitation of laminar speleothem (Fig. 1c).

Three bovid teeth (TNH2-10/CC10, TNH2-11/CC11, TNH2-12/CC12) recovered from the upper fossil-bearing breccia (LU2) were directly dated using coupled uranium series and electron spin resonance (US-ESR), providing a weighted mean age estimate of $151 \pm 37$ thousand years ago (kyr) (2-sigma) (Fig. 1b; Supplementary Tables 1, 2) and an age range of 188–117 kyr. Two large blocks of breccia (LCC1 and LCC2) from LU2 (upper) and one block of the silty clay unit (LCC3) from LU1 (lower) were removed for luminescence dating (Fig. 1b). These samples produced coeval age estimates of $143 \pm 24$ kyr (LCC1) and $133 \pm 19$ kyr (LCC2) for the deposition of the LU2 breccia and $248 \pm 31$ kyr (LCC3) for the underlying LU1 silty clay deposit (Supplementary Table 3). These ages are in stratigraphic agreement with the age of the overlying flowstone (CCF1), which was precipitated earlier than $104 \pm 27$ kyr based on the weighted mean of U-series age estimates on four separate sub-samples of flowstone carbonate (Supplementary Table 4). Bayesian modelling was performed on all independent age estimates to determine an overall geochronological framework for the site and tooth (Methods and Supplementary Fig. 3). The fossiliferous breccia including the tooth was deposited between 164 and 131 kyr (at 68% confidence limit).

**Fauna**. The Tam Ngu Hao 2 faunal assemblage comprises 186 identified dentognathic specimens (NISP) dominated by isolated teeth of large mammals, including several megaherbivores (Supplementary Table 5, Supplementary Data 2). Their analyses reveal typical taphonomic pathways of assemblages from karstic systems

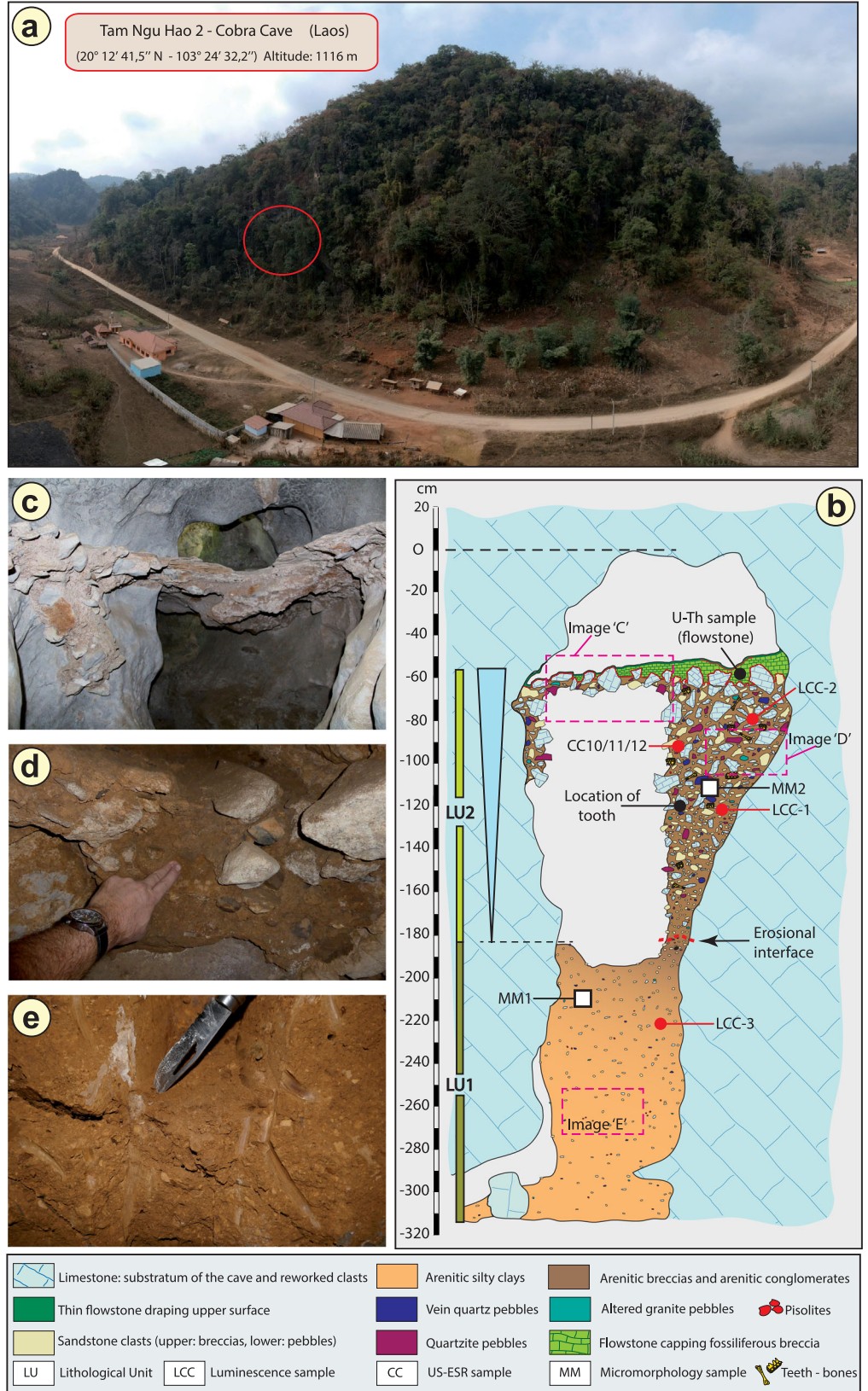

**Fig. 1 Geomorphological context and stratigraphy of TNH2. a** Aerial view of the site. The red circle indicates the entrance of Tam Ngu Hao 2 cave.
**b** Stratigraphy and sampling locations of the infilling of the cave, showing Lithological Unit 1 and 2 (LU1 and LU2) with the erosional interface between these layers indicated by a dashed red line; Micromorphological (microstratigraphic) samples (MM1 and MM2) are also shown. Encircled numbers denote approximate positions of photographs in **c**, **d** and **e**. **c** View of the flowstone capping the upper remaining part of LU2. **d** Detail of the arenitic breccia/conglomerate of LU2. **e** Detail of the arenitic silty clay of LU1.

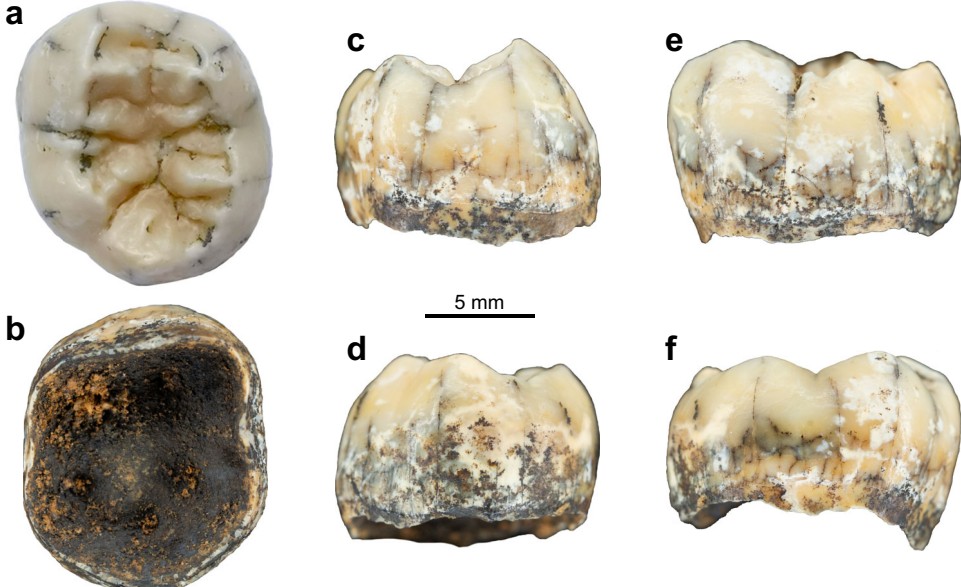

**Fig. 2 Views of the TNH2-1 specimen.** Pictures of TNH2-1 in occlusal (**a**), inferior (**b**), mesial (**c**), distal (**d**), buccal (**e**) and lingual (**f**) views.

in terms of representation of specimens and types of damage. Due to the energy associated with the deposition of LU2, only teeth of large mammals are present in the assemblage, and we note the absence of small and light teeth of any microvertebrates. More-over, most teeth are gnawed by porcupines, known to be a major accumulator agent in the region[24]. Therefore, the poor pre-servation of specimens precludes identification to the species level for most of the recorded taxa. The fauna bears close affinities to those known from the late Middle Pleistocene of southern China and northern Indochina and, to a lesser extent, Java, which is consistent with the sedimentary chronology of the site. It can be assigned to the "*Stegodon-Ailuropoda* faunal complex"[25–28]. We note the absence of Neogene taxa that persist in the Early Pleis-tocene and that of two key-species, *Pachycrocuta brevirostris* and *Gigantopithecus blacki*, which are good indicators of pre-300 kyr faunas in the region[26,27,29–31]. The archaic *Stegodon* persisted in Asia most likely until the end of the Late Pleistocene[32]. We recovered herbivores including *Tapirus*, *Stegodon*, and Rhino-cerotidae, that were adapted to canopied woodlands in the area. We also found animals such as the *Bos* species, small-sized Caprinae and large-sized Cervidae (possibly *Rusa unicolor*), which are all known to exhibit a great variability in their feeding behaviour and preferred habitats, from closed and intermediate forests to open grassland[33].

**Ancient protein analyses**. The enamel from the TNH2-1 tooth specimen was analysed using nanoLC-MS/MS and the recently developed approach for ancient enamel proteomes[34]. The TNH2-1 proteome is composed of a common set of enamel-specific proteins, all of which have previously been observed in Pleisto-cene enamel proteomes[34–36] (Supplementary Table 6). The enamel proteome has elevated levels of diagenetic protein mod-ifications (Supplementary Fig. 4a–d, Supplementary Table 7) and preserves serine (S) phosphorylation within the S-x-E motif previously observed in ancient dental enamel[34,35] (Supplementary Fig. 4e). Based on proteome composition and modification, as well as the absence of peptides matching any of these proteins in our extraction and mass spectrometry blanks, we consider our proteomic data as indicative of endogenous proteins deriving from the sampled enamel.

Unfortunately, no high-confidence peptides overlapped diag-nostic amino acid positions with sequence differences between *H. sapiens*, Denisovans, or Neanderthals, making further taxonomic assignment based on palaeoproteomics impossible. This is in line with previous research, which indicated that closely related hominin populations can be distinguished based on dentine and bone proteomes, while enamel proteomes are less informative in the context of close phylogenetic proximity[35]. Nevertheless, by comparing the sequences recovered from the TNH2-1 enamel proteome with that of extant hominids for which protein sequences are available, we find that the specimen belongs to a member of the genus *Homo* (Supplementary Table 8).

The absence of peptides specific to male-diagnostic amelogenin Y (AMELY) suggests that either the sampled molar was from a female individual or that AMELY-specific peptides were not observed due to degradation beyond the limit of detection of the instrument.

**External and internal structural analyses of the tooth**. Exter-nally, the TNH2-1 crown displays a coarse wrinkling pattern that is found in Pleistocene *Homo* (*H. erectus* s.l., European and Asian Middle Pleistocene *Homo* and Neanderthals), but is rare in modern *H. sapiens*. The mid-trigonid crest is well developed as commonly recorded in European Middle Pleistocene *Homo* and Neanderthals, while it is generally absent or less frequent in *H. erectus* s.l. and fossil and extant *H. sapiens*[37]. Below the external surface, the enamel-dentine junction (EDJ) of the tooth shows the dentine horns of the five main cusps and of a tuberculum intermedium and a low but uninterrupted mid-trigonid crest (Figs. 2 and 3, Supplementary Fig. 2; Methods). The latter feature is generally found in Neanderthals (80–100% depending on the molar position)[38–40] but is less frequent in *H. erectus* s.l. and *H. sapiens*[41–47] (Supplementary Fig. 5). In addition, the EDJ of TNH2-1 shows an internally-positioned metaconid reminiscent of Neanderthal molars[40] and a low crown topography similar to that of *H. erectus*[41–47]. These features, as well as a slight buccal shelf present on the EDJ of TNH2-1, are all expressed on the EDJ of the Denisovan molars from Baishiya Karst Cave (Xiahe, Gansu, China) (Supplementary Fig. 5)[15]. TNH2-1 dentine differs from the much higher and proportionally more mesiodistally com-pressed EDJ of Neanderthals and *H. sapiens*[39,40], as well as from

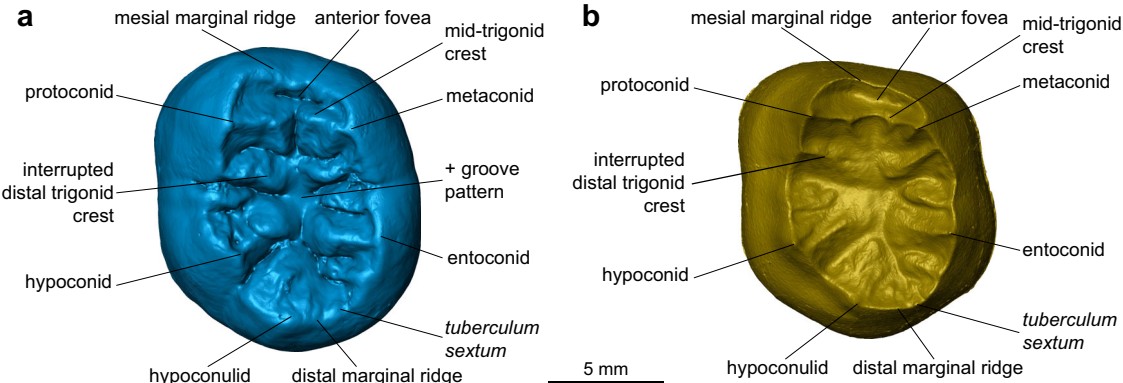

**Fig. 3 Morphological features on the virtual rendering of the TNH2-1 specimen.** Virtual renderings of the outer enamel surface (**a**) and enamel-dentine junction (**b**) in occlusal view showing the main morphological features.

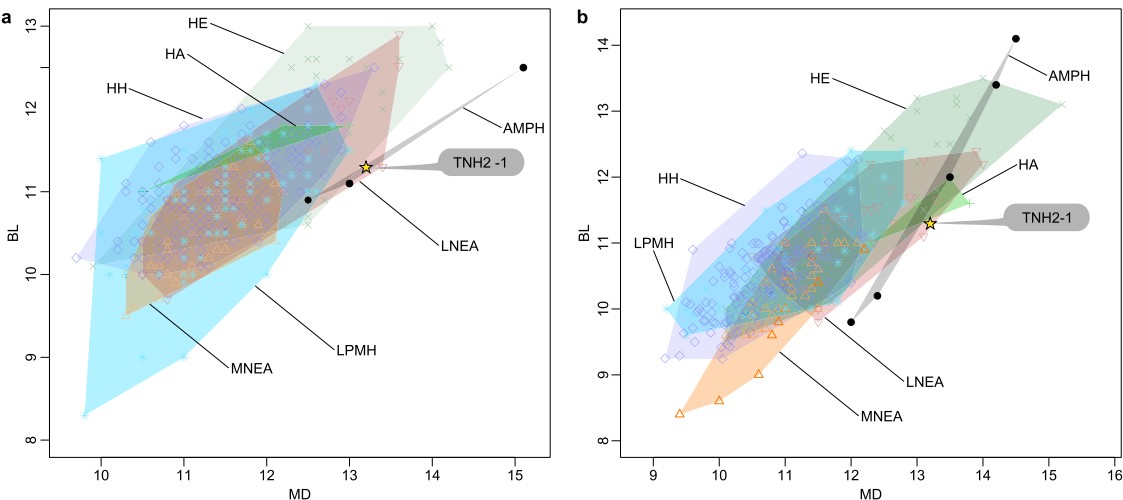

**Fig. 4 Metrical features of the TNH2-1 specimen.** Bivariate scatter plots of the mesiodistal and buccolingual crown dimensions of TNH2-1 compared with the M1s (**a**) and M2s (**b**) of *H. erectus* (HE; green crosses), *H. antecessor* (HA; green pluses), Middle Pleistocene Neanderthals (MNEA; orange triangles), Late Pleistocene Neanderthals (LNEA; red inverted triangles), Asian Middle Pleistocene *Homo* (AMPH; dark dots), Late Pleistocene modern humans (LPMH; blue asteriks) and Holocene humans (HH; blue diamonds). Source data are provided as a Source Data file.

the shorter dentine horns and more densely wrinkled occlusal basin of *H. erectus* s.l.[41–47] (Supplementary Fig. 5).

In terms of absolute dimensions, only Asian Middle Pleistocene *Homo* have larger tooth crowns than TNH2-1 (Supplementary Tables 9, 10). TNH2-1 crown metrics are within the ranges of variation for *H. erectus* s.l., *H. antecessor*, Asian Middle Pleistocene *Homo* and Neanderthals, but they statistically differ from the smaller crowns of European Middle Pleistocene *Homo* and from Pleistocene and Holocene *H. sapiens* (Fig. 4; Supplementary Tables 10, 11). With respect to tooth crown tissue proportions, TNH2-1 has a high percentage of crown dentine (Vcdp/Vc: 55.37%) with moderately thick enamel as shown by absolute and relative enamel thickness values (3D AET: 1.18 mm; 3D RET: 17.00; Supplementary Table 12). These crown tissue proportions match to those of the nearly unworn M2 of the Xiahe mandible[15] (Vcdp/Vc: 54.62%; 3D AET: 1.47 mm; 3D RET: 18.97) and the upper molar of Denisova 4 (3D RET: 15.27; B. Viola, pers. comm.), but within the ranges of variation of all comparative fossil and extant human groups (Supplementary Fig. 6a–c; Supplementary Tables 12, 13). Three-dimensional maps of topographic enamel thickness distribution show that TNH2-1 has the thickest enamel at the top of the hypoconid and hypoconulid cusps and in the distobuccal quarter of the crown (Supplementary Fig. 6d). In comparison, all other samples tend to

have the thickest enamel distributed on all buccal cusps and more spread on the buccal aspect of the crown, even if variable between groups and between molar positions. The M2 of the Xiahe specimen shows thicker enamel spread along the buccal crown aspect but its distribution pattern is partly obliterated by occlusal wear.

The EDJ shape of TNH2-1 was quantitatively compared with those of Pleistocene and Holocene human groups using geometric morphometrics (Methods). Landmark-based and surface deformation-based approaches were used, with both methods similarly distinguishing between *H. erectus* s.l., European Middle Pleistocene *Homo* and Neanderthals and *H. sapiens* using canonical variate and between-group principal component analyses (Fig. 5, Supplementary Fig. 7). Along CV2 and bgPC1, the higher EDJ and more externally set dentine horns of Neanderthals and *H. sapiens* are discriminated from the lower and more centrally positioned dentine horns of *H. erectus* molars. The CV1 and bgPC2 axes separate Neanderthals from modern humans, with the former having more internally placed mesial dentine horns and a more developed hypoconulid than the latter. TNH2-1 falls outside the ranges of all other groups. It has an intermediate EDJ shape between the low crown of *H. erectus* (but exceeding the variation of the latter group along CV1 and bgPC2) and the cusp position of Neanderthal molars (even if outside their

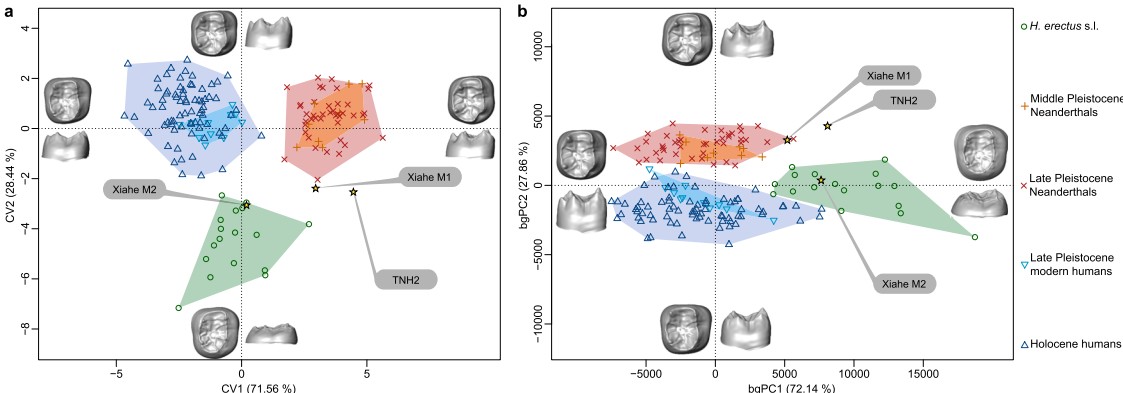

**Fig. 5 Geometric morphometric analyses of the EDJ.** Canonical variate analysis (**a**) and between-group principal component analysis (**b**) of the EDJ deformation-based shape comparison of TNH2-1, *H. erectus* s.l., the Denisovan specimen from Xiahe, Neanderthals and *H. sapiens*. Source data are provided as a Source Data file.

range of variation along CV2 and bgPC1). TNH2-1's closest morphological affinity lies with the Denisovan specimen Xiahe, which also displays Neanderthal-like features (Fig. 5, Supplementary Fig. 7).

### Discussion

Reconstructing dispersals and ultimately evolutionary trajectories of *Homo* in Asia depends on a currently poor fossil record. The Asian late Middle Pleistocene fossil record is mostly limited to the eastern part of the continent[4,8–10,15,48]. Any additional human remains from this time period documenting the evolution of *Homo* in southern Asia might thus help confirm previous hypotheses or reveal new lineages. Proteomic analysis of the TNH2-1 molar indicates that it most probably belongs to a female individual of the genus *Homo*. Morphometric analyses of the external and internal crown structural organisation allow us to reject a number of hypotheses regarding species assignment. TNH2-1 has large crown dimensions and a complex occlusal surface that differentiates it from the smaller and morphologically simpler teeth of *H. floresiensis*[49], *H. luzonensis*[50] and *H. sapiens* (Fig. 4). The EDJ shape shows a mixture of Neanderthal-like and *H. erectus*-like features, closely resembling the M1 morphology of the Denisovan specimen from Xiahe (Fig. 5, Supplementary Fig. 5). The similarities between TNH2-1 and *H. erectus* are mostly related to the proportionally lower crown, although *H. erectus* molars display even lower molar crowns and a narrower occlusal basin (Figs. 2 and 3, Supplementary Fig. 5). The Lao fossil shows clear Neanderthal-like features such as a well-developed mid-trigonid crest and internally-positioned mesial dentine horns, but differs with its much lower EDJ topography and occlusal basin shape.

The differences from Neanderthals that we observe do not preclude TNH2-1 from belonging to this taxon and would make it the south-eastern-most Neanderthal fossil ever discovered. However, considering the morphological particularities of TNH2-1 in unison, as well as the high-degree of morphodimensional similarities with the molars of the Denisovan specimen from Xiahe, the most parsimonious hypothesis is that TNH2-1 belongs to this sister group of Neanderthals. If TNH2-1 indeed belongs to a Denisovan, this occurrence, along with the recent discovery of a Denisovan mandible from the Tibetan Plateau, a high-altitude, hypoxic environment[15], would suggest that this Pleistocene Asian population possessed a high degree of plasticity to adapt to very diverse environments[51]. Available Denisovan dental remains indicate a mixture of traits consistent with the current paleogenetic evidence that Denisovans and Neanderthals are sister taxa[13,14,51–53] and are therefore expected to share some

craniodental features[15,54]. This is further supported by recent analyses that identified possible Denisovan skeletal characteristics based on unidirectional methylation changes including traits that have been linked to Chinese fossils such as Xujiayao and Xuchang[9,54]. Denisovans are notable for their large dentition, with some Neanderthal-like crown features[15,48,54], as well as distinctive cusp and root morphology[14–16]. In the absence of molecular analyses, looking for these combined features in the Asian human fossil record, including in fossils like the Penghu 1 mandible from the Taiwan Strait[55], may help identify more Denisovan specimens (Supplementary Fig. 8).

The alternative hypothesis that TNH2-1 belongs to a group of Neanderthals that made an incursion into southeast Asia (see for example discussions on fossils that may demonstrate this dispersal from Maba and Dali)[56,57] cannot be outright rejected, even if it is less likely.

The tooth from Tam Ngu Hao 2 Cave in Laos thus provides direct evidence of a most likely Denisovan female individual with associated fauna in mainland Southeast Asia by 164-131 kyr. This discovery further attests that this region was a hotspot of diversity for the genus *Homo* (Supplementary Fig. 8), with the presence of at least five late Middle to Late Pleistocene species: *H. erectus*[58], Denisovans/Neanderthals, *H. floresiensis*[49], *H. luzonensis*[50] and *H. sapiens*[20–22].

### Methods

Fossiliferous breccia has been extracted mecanicaly from the walls of the cave and some of the hardest blocks containing the fauna have been dissolved within a 10% formic acid solution.

Excavation, collection, export of fossils for study have been granted by the Ministry of Information, Culture and Tourism of Lao PDR in January 2019 with the support of the local authorities of Xon district, Hua Pan Province and the villagers of Long Gua Pa village. Results of this study will be shared with the authorities of Xon District and with the villagers of Long Gua Pa village, following the same information routine we set up at the beginning of each of our annual fieldworks.

**Microstratigraphic analysis**. For the purposes of analysing the microstratigraphy of the Cobra Cave sediments, an intact sediment block was removed from LU1 (MM1) and LU2 (MM2) (Fig. 1b). The unconsolidated sample (MM1, LU1) was extracted using a gypsum-impregnated bandage to stabilise the sediment. It was possible to remove the sample from LU2 (MM2) using a hammer and chisel because of the well-cemented nature of the breccia. The sample from LU2 was located in an area close (~30 cm) to the hominin tooth find-spot. Thin sections were prepared at Adelaide Petrographics (Adelaide, Australia). The block from LU2 did not require stabilisation and was cut using a rock saw. A ~100 μm slice was mounted on a glass slide and ground and polished down to a final thickness of 30 μm. The sediment block from the lower unconsolidated unit was impregnated with resin, following procedures described elsewhere[59,60]. Diagnostic features observed using a polarising microscope were recorded for each layer using standard protocols[61].

Micromorphological (microstratigraphic) analyses of cave sedimentary infills can elucidate the processes responsible for the formation of a fossil site, the depositional and post-depositional environments, and the context of fossil assemblages[59,60,62–64]. Micro-contextualisation can provide critical information regarding the context of fossils and how far they may have travelled from their original positions.

*LU1: Lower sands and silts (Supplementary Fig. 9a).* The lower sediments of LU1 appeared sandy and gritty in the field, but microstratigraphic analyses show that the sand-sized inclusions comprise silty clay aggregate grains of variable lithological composition. These lower sediments are generally loose but with partial cementation in localised zones. In broad terms, the sediments of LU1 (Supplementary Fig. 9a) consist of i) a matrix material of fine quartz sand in a speckled b-fabric and ii) large, coarse, sand-sized, rounded to well-rounded aggregate grains with variable colour, lithology and fabric arrangement, suggesting polygenetic (intra and extra-karstic) origins.

Groundmass ranges from single-spaced to double-spaced enaulic. At the macro-scale, the compound grains form the coarse fraction of the overall sediment, ranging from very fine gravel/very coarse to fine sand and in shape from angular to well rounded. The coarse component of the matrix material comprises very fine quartz sand and coarse silt, sub-angular to sub-rounded. Many of these aggregate grains exhibit internal stratification with micro-banding of finer and coarser material (primarily clay and silt), and grains vary in colour from pale brown to very dark brown. The sediments are moderately porous, exhibiting frequent planar, vughy and vesicle voids. Planar voids generally link vertically or sub-vertically across vughs and vesicles and are common adjacent to the edges of large compound grains. Stratified compound grains are generally much less porous and are often clay-enriched. Pedofeatures are rare in LU1 and are most commonly evident in the form of clay pendent coatings surrounding rounded to sub-rounded compound grains. Small ferrous particles are present in small quantities. Organic material is absent from this layer.

Field observations and micromorphological analyses together suggest that LU1 was deposited in a low to moderate energy hydrological environment. The coarse nature of the aggregate grains (very fine gravel, very coarse sand and ranging down to fine sand) suggest that water flowing through the karstic system reworked older cave fills adhered to the walls and roof higher up the cave catchment system (cf.[62]). The variable lithologies of these compound grains are consistent with reworking of fine-grained material from multiple sources, most likely of both intra- and extra-karstic origin. The very dark coloration of some (≤5%) of the aggregate grains suggest either reworked pedogenic material or pelite sourced from outside the site. Following deposition of these fossil-sterile sediments, minor post-depositional disturbance of the sediments occurred, evidenced by the well-developed void structure consistent with bioturbation. Bioturbation has contributed to the removal of any banding, laminations or structures diagnostic of specific fluvial sedimentation processes.

*LU2: Upper fossiliferous conglomerate/breccia (Supplementary Fig. 9b–t).* As stated above, the conglomerate was sampled to avoid gravel clasts and so this description pertains to the fine sediment matrix. Randomly distributed void spaces are present, with common very small vughs and planar voids, which are locally larger and more complex, sometimes forming chamber and channel voids. All void spaces are infilled with calcium carbonate to cement the sediment. Groundmass is generally a double-spaced porphyric c/f distribution pattern with frequent angular to sub-angular mineral grains (mainly quartz), small angular fragments of bone, large tooth fragments and some small (fine to medium sand-size) iron-stained nodules. Fabric is infrequently granostriated around coarse inclusions (e.g., quartz or rock grains). Pedofeatures are rare in the TNH2 cave sediments but infrequent clay coatings are recorded in some areas of the examined thin section, with thin coatings on some bone fragments. There are also regions where the sediment matrix is clay-enriched, probably the result of illuviation. Bone fragments are typically medium to coarse sand-size in thin section (with larger elements recovered from the sediment) and exhibit only moderate signs of chemical weathering, with low birefringence (in xpl) and some signs of oxide staining probably related to microbial activity. In some regions of the sample, small bone fragments are particularly weathered and exhibit small cracks and heavy oxidation, and these might therefore represent older, lag deposits. Large tooth fragments (~15 mm in size) are present, including both dentine and cementum in a well-preserved state.

The sediments of LU2 comprise fine to medium gravels, coarsening up to large gravel clasts and occasional cobbles in towards the upper capping flowstone. The gravel is suspended in a fine-grained matrix of silts and fine to medium sands, with variable quantities of clay. These sediments were deposited in a high energy environment, probably during a flood event or reconfiguration of the karst hydrogeological system. This probably occurred as a single event in the course of hours or days as we do not identify sub-units, potentially as both fluvial discharge and saturated debris flows. Mass movements are recorded in the microstratigraphy as granostriated b-fabrics surrounding coarse mineral grains, evidence that (rotational) pressure has been applied around grains. These sediments were probably water saturated and plastic, with percolating water translocating fine clays through the profile concentrating these fines in localised regions of the matrix, as recorded in localized regions of the sampled sediments.

Micromorphological analyses confirm field observations that small bone fragments and frequent large tooth fragments are common throughout the sampled area of LU2, indicative of a long residence time in the karstic network. The bone fragments are commonly angular retaining aspects of their original morphology and are generally well-preserved with the exception of moderate iron staining consistent with microbial action. In terms of the diagenetic and taphonomic signature recorded in these sediments, there is scarce evidence of acid etching or edge rounding of the organic components, although some quartz grains do exhibit weathering features. Only very few bone fragments (<5%) show signs of significant weathering consistent with longer storage time in the karstic network. Although we record small-scale bioturbation of the sediments, this is minor, and the void spaces created are not of sufficient size to be the source of mixing of the sediments.

**Dating**

*Luminescence dating of breccia matrix.* Two large blocks of the hard-cemented, fossil-bearing breccia (LCC1 and LCC2) from LU2 and one from the lower silty clay unit (LCC3) in LU1 were cut from the sections in situ and wrapped in black plastic (Fig. 1b). Within subdued red-light conditions, the light-exposed, outer layer was removed using a chisel and hammer and was retained as the dosimetry sample. These layers were broken up using a pestle and mortar and oven dried, then the entire fraction was milled and used for environmental dose estimation. The unexposed inner core was also broken up using a pestle and mortar and was processed using the standard sample purification procedures for feldspar separation[65] including a 10% wash in hydrofluoric acid for 10 min to remove the external alpha-dosed rinds[58]. All luminescence analysis was conducted at the 'Traps' luminescence dating facility at Macquarie University in Sydney, Australia.

Individual 180–212 μm feldspar grains were mounted onto coated, single grain discs in a $10 \times 10$ grid. The discs were loaded onto a carousel and processed in a Riso TL-DA-20 containing an automated Detection and Stimulation Head set up with a Dual laser single grain attachment with a Blue/UV sensitive Electron Tube PMT (PDM9107Q-AP-TTL-03) with maximum detection efficiency between 200 and 400 nm. The filters in the automated detection changer were set on the blue filter pack (Schott BG-39 and Corning 7–59 filters to transmit wavelengths of 320–480 nm[66]). The grains were stimulated using an IR (830 nm) 140 mW TTL modulated laser with a 3 mm RG-780 longpass filter (mounted directly in front of the IR laser), and the emissions were detected using the blue filter combination described above. The laser stimulated the grains for 2.5 s, first at 50 °C and secondly at 270 °C, after a 300 °C preheat according to the procedures of the pIR-IRSL protocol[67] (see *Procedural tests* below). The single-grain disc locating process was programmed to occur before any disc heating procedure to ensure that each grain received the same heating within the SAR cycles.

Equivalent doses were corrected according to the results of the anomalous fading tests (using a weighted mean fading rate of $2.0 \pm 0.2\%$ per decade) then run through a minimum age model (MAM) to identify the population that had the most bleaching prior to burial.

To obtain an estimate of the environmental dose rate for each of the samples, we first measured beta dose rates using a Geiger-Muller multi-counter beta counting of dried and powdered sediment samples[68] in the laboratory. Allowance was made for the effect of sample moisture content, different grain sizes[69] and HF etching on attenuation of the beta dose, and the total beta dose-rate contribution was calculated by comparing the beta count rate to a standard beta source (SHAP with a dose rate of 5.99 Gy/kyr) and magnesium oxide as a non-beta emitting background material. Secondly, thick source alpha counting using a Daybreak 583 intelligent alpha counter was used to obtain estimates of Uranium and Thorium[70] to estimate the gamma dose rate, and thirdly the difference between beta and alpha counting was used to estimate potassium values. These estimates were then converted to gamma dose rates using the conversion factors of Guerin et al[71]. Allowance was made for the effect of sample moisture content[65] on the external beta and gamma dose rates using a long-term water content of between $3 \pm 0.2$–$5 \pm 2\%$, which is similar to the measured (field) water content of between 3% and 7% and allows for an initial period of saturation when first deposited in the karst environment.

The total dose rate was then calculated using an effective internal beta dose rate of 0.84 Gy kyr$^{-1}$ [72,73] for the 180–212 μm feldspar samples (due to the radioactive decay of $^{40}K$ and $^{87}Rb$), which were made assuming K ($12.5 \pm 0.5\%$) and $^{87}Rb$ ($400 \pm 100$ μg g$^{-1}$) concentrations and included in the total dose rate. Cosmic-ray dose rates were estimated from published relationships[74], making allowance for the thickness of limestone above the cave (~20 m with an assumed density of 1.2 g/cm$^3$), sediment overburden at the sample locality (0.3–1.3 m with an assumed density of 2.0 g/cm$^3$), the altitude (~1116 m above sea level) and geographic latitude and longitude (20°S and 103°E) of the sampling site.

In addition, this estimation of dose rates was supported by high resolution gamma spectrometry to measure the activities of radionuclides in the $^{238}U$, $^{235}U$, $^{232}Th$ decay chains and of $^{40}K$ at the SGS laboratory in Melbourne. These activities were converted to beta and gamma dose rates using the conversion factors of Guerin et al.[71] and adjusted for the same long term water content as mentioned above. These activities provide an alternate dosimetry and were also used to estimate the potential disequilibrium in this cave environment.

*Procedural tests.* Previous luminescence analyses at a neighbouring site, Tam Pà Ling[20–22], revealed high dose rate (~3–4 Gy/kyr), a saturating quartz signal with depth >4 m and feldspars with a high dose response and stable signal that could be isolated from the unstable fading signal[22]. Building on this experience, we focused on just the feldspars and applied the same post-infra-red infra-red-stimulated-luminescence (pIR-IRSL) protocol to overcome the problems of anomalous fading[75]. However, due to the number of feldspar grains used on each disc, which produced only maximum ages for sediment deposition, combined with a low yield in potassium feldspar grains at these sites, we decided to apply a single-grain pIR–IRSL technique rather than the single aliquot technique.

We applied the same procedural tests described in Shackelford et al.[22] but modified them for single-grains of feldspar from sample LCC1 (as this sample contained slightly more feldspar grains than the other two) in line with Roberts[76] and Rhodes[77] protocols (Supplementary Table 14): 1) a preheat plateau test; 2) fading tests; 3) bleaching tests; and 4) dose recovery tests. Due to the small amount of grains in the 180–212 µm size fraction, we saved this for the $D_e$ estimation and instead used the smaller 90–180 µm size fraction (and corresponding 200 µm single-grain discs) for all the procedural tests using the following preheat and IR stimulation combinations: 1) 250 and 225 °C (pIR-IRSL$_{50,225}$)[67,78]; 2) 280 and 250 °C (pIR-IRSL$_{50,250}$)[79]; 3) 300 and 270 °C (pIR-IRSL$_{50,270}$)[79]; 4) 320 and 290 °C (pIR-IRSL$_{50,290}$)[80,81]; and as the expected De is >450 Gy, we also tested 5) 320 and 200/290 °C (pIR-IRSL$_{200,290}$)[82]. As many of the discs had very few feldspar decays, we used 4 single grains discs with a total of 16 grains and recycled these grains for all the tests. This is not ideal, and the use of fresh grains would have been preferred but the low feldspar yield meant that we had to prioritise the remaining sample for the actual measurements. All tests mostly followed the procedures outlined in Shackelford et al.,[22] with the following differences:

1. Preheat plateau and dose recovery tests—the different preheat/IR laser stimulation combination was applied to each SG disc and to a different number of grains depending on how many grains luminesced on each disc. We then added a surrogate dose of 20 Gy to try and recover this known dose from these discs.
2. Bleaching tests—the same 4 discs had a dose of 20 Gy applied and were bleached in a solar simulator and measured using the same preheat/IR laser stimulation combinations.
3. Following the protocols of Rhodes[77] for testing anomalous fading in SG of feldspars, we employed 1) individual single-grain fading using the 4 discs for multiple delay times; 2) multiple grain measurements by using the SG discs as single-aliquots; and 3) the use of a standard fading value assessment by the degree of saturation for an infinite aged sample derived from the weathered granite pebbles found in the deposit and analysed using small single-aliquots.

The tests revealed that the flattest plateau was provided by the pIR-IRSL$_{50,270}$, pIR-IRSL$_{50,290}$ and pIR-IRSL$_{200,290}$ signals (Supplementary Fig. 10c), while the pIR-IRSL$_{50,270}$ signal provided the best recovery of the surrogate dose (with a dose recovery ratio of 0.995) and lowest residual values after bleaching (<10 Gy) (although calculated, these residual doses were not subtracted from the $D_e$) (Supplementary Fig. 10e). The pIR-IRSL$_{50,290}$ and pIR-IRSL$_{200,290}$ signals produced the largest residuals doses, which raises concerns over signal overestimation in this cave environment where there is a strong likelihood for partial bleaching. The fading tests from the single aliquot tests were much lower than from the single-grain tests, which revealed a range of fading from the highest IR$_{50}$ to the lowest pIR-IRSL$_{50,270}$, pIR-IRSL$_{50,290}$ and pIR-IRSL$_{200,290}$ signals (Supplementary Fig. 10d). We also observed a variation in fading rates among the single-grains (ranging from negative values to higher values that would not allow for any natural signal, with an average over the 16 grains of 2.3% per decade), but this was independent of the grains' sensitivity. When using the pIR-IRSL$_{50,270}$ signal on the single-aliquot discs, this g value was reduced to an average of 1.7% per decade. We decided to use a mean fading rate of 2.0 ± 0.2% per decade as a mid-point before the two techniques. Surprisingly the pIR-IRSL$_{200,290}$ signal displayed the same amount of fading as seen in the pIR-IRSL$_{50,270}$ and pIR-IRSL$_{50,290}$ signals, indicating no apparent advantage in a higher initial temperature for IR stimulation in these samples.

The weathered granitic cobble provides the closest representation of the provenance of the feldspar grains from LCC1 and LCC2. Following the procedures of Brown et al.[83], we used 12 very small single-aliquots (as the signal was too insensitive to use single-grains) and added large doses of 200 and 400 Gy to the natural signal (N + 200 Gy and N + 400 Gy) for eight discs and zero dose to the final four discs. All discs were measured using the ph/stimulation temperature combinations described above (Supplementary Fig. 10f). In all the discs, the IR$_{50}$ signal grows with dose as does the pIR-IRSL$_{50,225}$ signal to a lesser degree, but the pIR-IRSL$_{50,290}$ and pIR-IRSL$_{200,290}$ signals change by only a small amount. This indicates that the field saturation is close to the laboratory saturation using these signals. However, to be conservative we used a fading rate of 2.0% per decade. We used the fading correction method of Huntley and Lamothe[84], which provides an estimation of the fading for small fading rates although it is designed for the linear portion of the growth curve.

As these feldspars contain very large De values close to the point of saturation, we conducted saturation tests on the same weathered cobble mentioned above[83].

We first ran the sample as a normal pIR-IRSL$_{50,270}$ to observe the dose response of a saturated sample to ensure that the fitting procedure could accommodate these high dosing values.

*Comments on the luminescence dating of the breccia.* The feldspars had low sensitivities yielding very low acceptance rates; in all over 8,300 grains were processed with only 96 accepted grains (1.16% acceptance rate). The 180–212 size fraction was chosen for single-grain analyses as this was the largest size fraction available; the 90–125 size fraction yielded negligible feldspar grains. Rejections were based on the protocol of Jacobs et al.[85] (see Supplementary Table 15). The grains were tested using XRD to verify that they were in fact K-rich feldspar grains, and the results indicated that they contained slightly lower than expected K contents (between 10.9 and 12.5%). Most of the grains had no decay whatsoever. These results were challenging considering the small amounts of sample yield meant that we were unable to get the number of accepted grains to a statistically meaningful number before running out of sample (LCC1- 48 grains, LCC2 - 25 grains and LCC3 - 23 grains). However, the value of the single-grain approach is that we can isolate individual dose populations rather than an average of many populations that result in a maximum age. Thus, the age estimate derived from the single-grain method (even with a small number of grains) is much closer to the true burial age of the sediments than a multiple-grain, single-aliquot approach.

The sedimentological and microsedimentological analyses provide clues as to the origins of the feldspar grains. The thin sections reveal a very low percentage of feldspar grains in the matrix, and this was confirmed by an analysis of the breccia itself, which contains minimal granitic pebbles. However, the few granitic pebbles found were highly weathered and could be the extra-karstic source for the K feldspar grains found in the samples. This also explains the low luminescence sensitivity of the grains. As for the bleaching potential of the grains - an extra-karstic origin bodes well for bleaching, but this depends on when the pebbles were weathered and released the K feldspar grains into the matrix. If weathered in situ, the chances for bleaching are reduced. This, combined with a rapid flooding event in a karst environment, is not the best scenario for bleaching grains. A single-grain approach in this situation is well justified to identify the grains that are the most bleached, however there is a chance that none of the grains have been bleached. In this situation even the apparently well bleached grains can still be overestimating the burial age, meaning that all the grains are effectively maximum ages. But when taken in the context of the relationship between the sediments and fossils, it is clear that the deposition of the sediments is the last event to occur as the flood waters would have gathered the fossils from the landscape (or incorporated already existing fossils from within the karst environment), resulting in the sediment ages being the youngest—even if they are maximum ages. The age of the sediments does not impact on the age of the fossils, which are all consistently older and had all accumulated by at least 104 kyr (the age of the overlying flowstone).

The striking difference between the environmental dose rate of the LCC1 (3.6 Gy/kyr) and LCC2 (5.0 Gy/kyr) in the same breccia unit can be explained by the heterogeneous nature of the deposit. The inverse grading suggests a rapid single karstic flood event of increasing energy. The flood waters deposited the arenitic silty clays before depositing the clasts in a coarsening, upwards sequence. At the point of LCC1, the clasts:matrix ratio was ~50% and was dominated by micritic and spartic limestone clasts; this increased to ~80% at LCC2 and was dominated by the granitic, quartzite and vein quartz pebbles derived from the underlying geology within a radius of ~20 km from the cave. This large difference in clast percentage and composition is enough to explain the difference in environmental dose rates. However, to make sure the values were indeed accurate we applied high resolution gamma spectrometry to both blocks. This produced a coeval result for LCC1 and a slightly higher dose rate for LCC2 (see Supplementary Table 3).

We used the high-resolution gamma spectrometry data from samples LCC1 and LCC2 to further investigate the differences in dosimetry and to test whether the deposit is in secular equilibrium. High resolution gamma spectrometry of the cave sediments is particularly useful as it provides information on the entire $^{238}U$ and $^{232}Th$ decay chains; therefore, the concentrations of the daughter as well as the parent nuclides can be explored. In this sedimentary environment, three potential disequilibrium states have been identified: 1) in the $^{238}U$ chain a deficiency of 16% in $^{226}Ra$ compared to $^{238}U$; 2) a $^{210}Pb$ a deficiency of between 17 and 22% compared to $^{226}Ra$; and 3) in the $^{232}Th$ chain, a small excess of 7% in $^{228}Th$ compared to $^{228}Ra$. This disequilibrium is more pronounced in sample LCC2 than displayed in the sediments around LCC1. This is another example of the differences between these areas, which must again be attributed to the difference in clasts and composition and the capacity of the more clast-supported sediments in LCC2 to allow water seepage, the leaching of the more mobile radionuclides such as uranium and radon loss. Thorium is thought to be immobile in most sediments[86], and the short half-lives (t) of the daughter nuclides ($^{228}Ra$ = 5.7 yrs and $^{228}Th$ = 1.91 yrs) and $^{210}Pb$ (22 years) means that this effect of excesses in these radionuclides is negligible. The leaching of uranium appears to be more of a problem in LCC2 than LCC1, but a deficiency of 16% would only have a small effect on the total dose rate and has been incorporated into the error estimate for that sample. Both samples display a loss of radon as represented by a $^{210}Pb$ deficiency compared to $^{226}Ra$, but the loss is 5.45% greater in LCC2. This can also be attributed to the presence of larger clasts that allow radon loss through the more

matrix supported unit. However, radon loss does not explain why LCC2 has a much higher dose rate as the greater loss of radon in this sample should instead cause a decrease in dose rate. The limestone clasts could also be the source of the disequilibrium, which are more abundant in LCC2. As the mineralogical assemblage of the clay fraction does not vary between samples, the variation in $^{226}Ra$ deficiency could reflect the proportion of limestone clasts in the sedimentary layers. Despite all of these differences, the main largest difference in the environmental dose rate between the samples is an almost doubling of $^{40}K$ in LCC2. This could be derived from the higher potassium feldspar content from the weathered granitic cobbles.

The environmental dose rate of the breccia (3.6–5.0 Gy/kyr) is significantly higher than the lower silty unit at 2.7 Gy/kyr. Thus, the resulting breccia $D_e$ values were very high at between 400 and 2000 Gy with $D_0$ values of between 600 and 1000 Gy. This meant that some grains were close to the point of saturation (~1000–1200 Gy) and pushed the limits of feldspar dose saturation. This made the rejection criteria of utmost importance to ensure that the saturated grains were not included in the accepted grains. The feldspars obtained from the weathered cobble provides the best test of the saturation characteristics of these samples, as they have never been bleached so they should display an infinite age. The initial fading test using these grains displayed grains that saturate ~1500–2000 Gy with D0 ~600–1000 Gy. This is within the limits of our $D_e$ estimation.

Due to the problems associated with uranium leaching in cave environments we decided to use the environmental dose rates derived from the high-resolution gamma spectrometry as a more conservative approach. Using these dose rates, the three blocks produced age estimates that are consistent with their stratigraphic location and within their error margins (within 1 σ). The two breccia age estimates (LCC1 143 ± 24 and LCC2 133 ± 19 kyr) are statistically indistinguishable from each other despite having a large difference in dose rates (3.6 and 5.0 Gy/kyr, respectively). This suggests that the breccia matrix had indeed been deposited rapidly within one unit, as suggested by the sedimentological analyses. The underlying silty unit (LCC3 248 ± 31 kyr) with minimal clasts had a much lower dose rate and much lower feldspar grain acceptance rate (0.2 %), indicating a different composition and/or different depositional circumstances. All three samples contained a wide range of $D_e$ values with overdispersion values of between 23–38%. This justified the use of the MAM, which has identified the grains that were the most bleached prior to burial inside Tam Ngu Hao cave.

*Uranium series dating of flowstone.* U-series dating of the overlying flowstone at Tam Ngu Hao 2 provides a minimum age for the deposition of the breccia and associated fossils. Four separate sub-samples were drilled from the fresh cross-section of a hand specimen of the in situ flowstone using a hand-drill. Calcite was sampled from the in situ flowstone using a hammer and chisel. The powdered sub-samples were subjected to chemical treatment and isotopic measurements by mass spectrometry[87]. U-series dating of the speleothem samples was conducted in the Radiogenic Isotope Facility (The University of Queensland) using a Nu Plasma multi-collector inductively coupled mass spectrometer (MC-ICP-MS). Analytical procedures followed previous publications for MC-ICP-MS[87–89]. $^{230}Th/^{234}U$ ages were calculated using Isoplot EX 3.75[90] and half-lives of 75,690 years ($^{230}Th$) and 245,250 years ($^{234}U$)[91].

*Comments on the uranium series dating of the flowstone.* The overlying flowstone is composed of poor-quality calcite and has a high detrital component in all four sub-samples, as reflected by the high levels of $^{232}Th$ concentrations and very low $^{230}Th/^{232}Th$ activity ratios (0.84 to 1.0) that afforded imprecise and unreliable age results. This flowstone is at least younger than 153 ± 1 kyr, which is the youngest uncorrected age among all four sub-samples (i.e., the maximum age of the sub-sample). The non-radiogenic $^{230}Th$ correction using the standard Bulk-Earth $^{230}Th/^{232}Th$ activity value of 0.825 with an assumed uncertainty of 50% for non-radiogenic $^{230}Th$ correction has resulted in very large error magnifications, with one of the four sub-samples returning a negative age value, pointing clearly to over-correction of the non-radiogenic $^{230}Th$. A better estimate of the most reliable correction is based on the assumption that the real detrital $^{230}Th/^{232}Th$ activity ratio should be lower than the bulk-Earth value of 0.825. In order to obtain a better estimate, we plot the measured $^{234}U/^{232}Th$ activity ratios against the measured $^{230}Th/^{232}Th$ activity ratios, which defines a best-fit line with an intercept value of 0.5 on the $^{230}Th/^{232}Th$ axis. In other words, this intercept represents the common initial (or detrital) $^{230}Th/^{232}Th$ activity ratio of the four sub-samples. Thus, a new correction scheme using a non-radiogenic $^{230}Th/^{232}Th$ activity value of 0.5 is more appropriate. Based on this new scheme, the four sub-samples give indistinguishable corrected ages with a weighted mean of 104 ± 27 kyr (See Supplementary Table 4 for details). When trying to establish the age of the underlying breccia, this result as a standalone age estimate is not particularly useful, but as part of a chronological section it does provide a 'minimum' age estimate for the entire stratigraphic sequence of flowstone, breccia and underlying silty unit and has thus been used in the Bayesian model.

*Coupled uranium series and electron spin resonance (US-ESR).* Three bovid teeth (TNH2-10/CC10, TNH2-11/CC11, TNH2-12/CC12) recovered from the upper fossil-bearing breccia (LU2) were used for direct dating by coupled US-ESR

following the protocol of Joannes-Boyau et al.[92–94] (Fig. 1b). All three teeth were first sectioned in half using a benchtop high precision diamond saw then polished to a 50 μm smoothness to expose a flat surface of dentine and enamel for $^{238}U$ concentration screening using an ESI NW213 and ICPMS quadrupole Agilent 7700 for concentration. A small fragment of dentine and enamel of each faunal tooth was then removed using a 300 μm thick diamond blade attached to a hand-held rotary saw for U-series analyses using a LA-MC-ICPMS Thermo Neptune plus coupled to an ESI NW193 laser ablation unit (Supplementary Fig. 11). Isotopic ratios and concentrations were obtained by LA-MC-ICPMS and quadrupole ICPMS analyses, respectively. An average value was calculated for each dental tissue and used for the US-ESR model. Baseline and drift were corrected using a NIST 612 glass disc, while a fossil hippopotamus tooth of known U-series concentration was used to correct $^{234}U/^{238}U$ and $^{230}U/^{238}U$ ratios and assess the accuracy of measurements. Another enamel fragment from each tooth was used for ESR measurements. ESR dating was performed on a Freiberg MS5000 X-band spectrometer at 1 G modulation amplitude, 2 mW power, 100 G sweep, and 100KHz modulation frequency, coupled to a Freiberg X-ray irradiation chamber, which contains a Varian VF50 x-ray gun at a voltage of 40KV and 0.5 mA current[95,96]. Fitting procedures were carried out with the MCDoseE 2.0 software that uses a Bayesian framework approach where the solution is a full probability distribution on the dose equivalent[97].

The CC10, CC11 and CC12 enamel fragments removed from each sample and used for the ESR measurements were directly in contact with the dentine on one side and directly in contact with the sediment on the other side (no cement). The outer surface of the enamel (in contact with the surrounding sediment) and the dentine directly attached to it were removed using a diamond blade rotary tool. Simultaneously, 100 μm on each side was removed to avoid alpha particle contribution. Each tooth fragment was then cut in three separate pieces (A, B and C), with A (50 Gy) and B (200 Gy) to be pre-irradiated with one single gamma dose prior to the x-ray irradiation and C irradiated with x-ray solely. Fragments were mounted onto a teflon sample holder, allowing the fragment to be exposed directly to the x-ray source behind a 200 μm aluminium cover shielding. To estimate the ESR equivalent dose ($D_e$), each fragment was irradiated seven times, following exponentially increasing irradiation times (i.e., 90 s, 390 s, 975 s, 1891s, 3604 s, 7380 s and 12382 s with an average dose rate for CC10, CC11 and CC12 of 0.231 Gy/s, 0.203 Gy/s and 0.249 Gy/s, respectively). The x-ray emission received by the bovid teeth was calibrated using added known gamma irradiation doses on fragments A and B performed at the Australia's Nuclear Science and Technology Organisation (ANSTO). During each irradiation step, the output of the x-ray gun was recorded to allow an accurate determination of the dose received by the sample at each step. The dose response curves (DRCs) were obtained by averaging the peak-to-peak $T_1$-$B_2$ ESR intensities recorded for each irradiation dose over 180° measurements (20° step) and merged into a single spectrum[98]. Isotropic components and baseline corrections were applied uniformly across the measured spectra[94].

The direct dating of TNH2-1 was attempted but to minimise damages to the valuable sample, the superficial layer of residual dentine at the root vestige was used. Unfortunately, analyses have shown a high degree of diagenesis to the micro sample that completely overshadowed any meaningful interpretation of an age for the fossil.

*Comments on coupled uranium series and electron spin resonance (US-ESR).* A non-orientated components (NOCORs) of the ESR signal of 16%, 14 and 8% for TNH2-10/CC10, TNH2-11/CC11 and TNH2-12/CC12, respectively, was extracted using the Joannes-Boyau[94] method from the signal. The final $D_e$ values were obtained by fitting a single saturating exponential (SSE) through the ESR intensities and by selecting the appropriate maximum irradiation dose (Dmax) in order to avoid dose estimation inaccuracy[99]. All errors were calculated with a 2-sigma interval. Sediments surrounding the teeth were collected and analysed in solution by ICPMS after an acid (1:3 Nitric and HCL) digestion method. The external gamma dose rate was calculated using the sediment content measured by ICPMS and assuming a full 4π geometry. The total dose rates for TNH2-10/CC10, TNH2-11/CC11 and TNH2-12/CC12 were calculated at 1482 ± 224, 1253 ± 192 and 1187 ± 208, giving US-ESR ages of 164 ± 24 kyr, 149 ± 22 kyr and 140 ± 23 kyr, respectively, using a Monte Carlo approach[100] (Supplementary Table 2) and assuming equilibrium for the rest of the decay chain. However, the uranium diffusion fits the open-system model with accumulation of uranium at the enamel dentine junction and in cracks as well as presence of leaching of uranium and enrichment of thorium 232. For sample TNH2-10/CC10, the $^{230}Th/^{234}U$ ratio in the enamel had to be assumed (see Supplementary Table 2(*)) because of leaching and/or sediment enrichment (measured $^{230}Th/^{234}U$ ratio >1 and concentration of U is low while thorium is high). Unfortunately, in order to model the age using the Monte Carlo calculation from Shao et al.[100], we used the same $^{230}Th/^{234}U$ ratio to that of the dentine for the enamel, introducing a sizable assumption into the age calculation of this particular sample. Under these conditions, the combined age estimate places the sequence slightly older than 150kyr. TNH2-10/CC10 remains slightly older, but within error of TNH2-11/CC11 and TNH2-12/CC12, and therefore statistically indistinguishable. Furthermore, the dose equivalent ($D_e$) of TNH2-10/CC10 was calculated to be 1.3 and 1.5 times greater than for TNH2-11/CC11 and TNH2-12/CC12, respectively (Supplementary Table 2). Additionally, the NOCORs concentration for

TNH2-12/CC12 might have been underestimated due to an unusually poor angular response of the fragment. A similar NOCOR content as for TNH2-10/CC10 or TNH2-11/CC11 would increase the $D_e$ by ~8%, which would bring the age closer to TNH2-11/CC11. The spatial distribution of the U-series results of each tooth gives a better understanding of the diffusion process within the dental tissues (see Supplementary Fig. 12a–c). Apart from punctual intrusion of sediments in cracks and other dental defects (Supplementary Fig. 12c), the distribution is characteristic of a diffusion pathway with uptake from pulp cavity into the dentine or to a lesser extent from surrounding sediments to the outer layer of the enamel in contact with the sediments. The age and concentration gradient observed with the distance moving away from the uptake origin is not reflected in the U-series ratio measured. Thus, potential leaching and/or late enrichment may have occurred but would be marginal with regards to the age calculation. This diffusion pattern reinforces the U-series age results and gives an average age estimate of $189 \pm 9$ kyr, $132 \pm 13$ kyr and $124 \pm 16$ kyr (2-sigma) for TNH2-10/CC10, TNH2-11/CC11 and TNH2-12/CC12, respectively. The p-value (diffusion function in dental tissues of uranium) in all three teeth is approaching $-1$ or in other words, equivalent to an early uptake of the isotopes into the teeth. This in turn explains why both U-series and US-ESR dating results are within error of each other. In summary, the US-ESR results of the faunal teeth provide a weighted mean age of $151 +/- 37$ kyr for the human remains. At last, while the fossil teeth were unearthed at a depth of 20 cm from the surface, the structure of the breccia indicates that a large part had eroded away in the recent time. Therefore, the external dose rate was modelled assuming a full 30 cm sphere of breccia surrounding each fauna sample. If we model the external dose rate to the current burial configuration, the external dose rate would amount to $699 +/- 121$ mGy/kyr. Because of the significant role of the external component to the total dose rate, the age estimates would be shifted to $190 +/- 28$ kyr, $177 +/- 26$ kyr and $168 +/- 27$ kyr for CC10, CC11 and CC12 respectively.

*Modelling of the breccia unit and associated fossils.* To evaluate the uncertainties of the integrated dating approach to the site (Supplementary Tables 1–4), Bayesian modelling was performed on all independent age estimates using the OxCal (version 4.2) software[101] available at https://c14.arch.ox.ac.uk/oxcal.html (Supplementary Fig. 3). The analyses incorporated the probability distributions of individual ages, constraints imposed by stratigraphic relationships and the reported minimum or maximum nature of some of the individual age estimates. Each individual age was included as a Gaussian distribution (with mean and s.d. defined by the age estimate and their associated uncertainties), and the resulting age ranges for each unit are presented at 1 σ.

No attempt was made to remove identified outliers. This is because we do not know the underlying 'true' age depth model, and we are using several different dating methods so it is difficult to specify the criterion to identify true outliers. Rather than this approach, we have explicitly specified minimum and maximum ages where appropriate to do so in keeping with the nature of the dating methods and the quality of the results. We believe this is a superior method compared to outlier analysis in this context, as it avoids unnecessary bias (e.g., in the choice of criterion) and represents a more conservative approach.

*Comments on modelling of the breccia unit and associated fossils.* The age estimates are coeval and the uncertainties are relatively small. As such, the identified boundary ages are not sensitive to removal of individual dates or to changes in, for example, the model calculation resolution. None of the changes we made to the model set-up produced appreciable differences in the age model results. The age of the breccia at Tam Ngu Hao 2 was conservatively estimated as the boundary between the overlying flowstone unit ($104 \pm 27$ kyr) and the age estimates of the breccia and fossils from within the breccia ($131 \pm 15$ kyr) and the boundary between the breccia and underlying silt unit ($164 \pm 17$ kyr), incorporating all of the constraints described above and the resulting age estimates (Supplementary Tables 1–4). The final modelled ages are presented in Supplementary Fig. 3 and the script presented in Supplementary Fig. 13. The age estimates from the breccia, derived from both US-ESR dating of teeth and pIR-IRSL dating of breccia matrix, are coeval within error margins despite constraining two different events, i.e., the death of the fauna and the deposition of the breccia unit, respectively. This indicates that the timing between death and final burial within the cave was very short, and that the breccia was deposited rapidly as one unit.

### Detailed morphological analysis of the tooth

*Description of TNH2-1.* TNH2-1 is an isolated mandibular left permanent molar crown germ with the root growth just initiated and reaching ~1.8 mm below the crown (Fig. 2a–f). The enamel is well-preserved, showing a shiny beige patina with whiter spots. There is a slight enamel hypoplasia toward the cervix and minute pits at mid-crown height on the buccal and mesial aspects, but no carious lesion. The dentine is dark brown on the external surface with some locally altered areas showing that the underlying dentine is yellower. The occlusal outline is subpentagonal. The absence of occlusal wear (stage 1[102]) and of interproximal mesial and distal facets indicates that the tooth was still unerupted at the time of death. There are five main cusps (protoconid>entoconid≥hypoconid>metaconid> hypoconulid) connecting in a + groove pattern in the middle of the occlusal basin, as well as a *tuberculum sextum* (C6) located on the hypoconulid distal marginal crest (Fig. 3). There is a prominent shoulder on the mesial crest of the hypoconid

that manifests as a small tubercle at the enamel surface. There is a high and continuous middle mid-trigonid crest (with a slight depression centrally) and an interrupted distal trigonid crest with a buccal segment initiating on the distal ridge of the protoconid and a lingual segment originating on the distal ridge of the metaconid (type A[103]). This trigonid crest pattern corresponds to a type 8 pattern of Martinez de Pinillos and collaborators classification[103], with a continuous mid-trigonid crest and a discontinuous distal trigonid crest arising from the distal segments. There are also shorter crests set distally to the distal trigonid crest and larger crests running from the talonid cusps down toward the centre of the occlusal basin. The anterior fovea is narrow, buccolingually extended and deep (type 4 according to ASUDAS[104,105]). There is no protostylid (type 0 according to ASUDAS[104,105]) although a faint buccal shelf is detected along the buccal face at the EDJ (Fig. 3b). At the EDJ, it can be observed that the mesial dentine horns are more centrally placed and more tilted toward the centre of the occlusal basin than the distal ones. In inferior view, five pulp horns are discernible despite the immature stage of the pulp chamber.

The overall pentagonal crown shape with subparallel buccal and lingual walls, the presence of a complete mid-trigonid crest, the well-expressed hypoconulid and C6, as well as the mesiodistally elongated diameter (see Figs. 2 and 3; Supplementary Fig. 2; Supplementary Tables 10 and 11) suggest that it more likely represents a first or second molar rather than a third molar. Statistical analyses performed on the EDJ shape (see Supplementary Table 16) classify TNH2-1 as an M2, but its marked resemblance with the M1 of the Xiahe mandible (as attested by the close position of both teeth in the GM analyses of the EDJ) and obvious difference with the M2 of the latter that shows an unusual occlusal morphology (with a marked buccolingual extension of the occlusal basin associated to the presence of a supernumerary root below; see ref. [15]) cast doubts about its metameric position (see Supplementary Fig. 5). The fact that the M1 and the M2 of the Xiahe specimens are both classified either as M1 (in the semilandmark analyses; Supplementary Table 16) or as M2 (in the DSM analyses; Supplementary Table 16) may also indicate a particularity of Denisovan molars. Indeed, while the unsecure attribution of the M2 is not surprising considering the above mentioned unusual EDJ shape, the M1 exhibits no anomaly and should be classified as a M1 in both analyses. As indicated by the difficulty to identify the Denisova 4 and 8 molars as M2s or M3s, this ambiguous result regarding the Xiahe molars may imply that Denisovan molar metameric variation differs from that of the groups included here as reference, preventing us from unambiguously attributing TNH2-1 to a M1 or to a M2 if it belongs to the latter group.

*Morphological analyses*

## X-ray microtomography

The TNH2-1 specimen was scanned using the X-ray microfocus instrument (X-µCT) diondo d3 at the Max Planck Institute for Evolutionary Anthropology, Leipzig, Germany. Acquisitions were performed according to the following parameters: 130 kV, 70 µA, 0.5 mm brass filter, 2970 images taken over 360° (~0.12° of angular step) with an averaging of 2 (frame averaging), integration time of 1000 ms. The final volumes were reconstructed with a voxel size of 9.14 µm. The microtomographic acquisitions of the comparative fossil and extant hominid specimens were performed using various equipments including X-µCT and synchrotron radiation (SRX-µCT) and reconstructed with voxel sizes ranging from 10.25 to 57.50 µm.

## Data processing

A semi-automatic threshold-based segmentation was carried out in Avizo 8.0 (FEI Visualization Sciences Group) following the half-maximum height method (HMH[106]) and the region of interest thresholding protocol (ROI-Tb[107]), taking repeated measurements on different slices of the virtual stack[108]. A volumetric reconstruction was then generated for each specimen.

## Crown dimensions

The mesiodistal and buccolingual tooth crown diameters of TNH2-1 were measured on the original specimen with Mitutoyo Digimatic calipers (to the nearest 0.1 mm) and then on the virtual surface generated after segmentation of the microtomographic record. These measurements were compared with the tooth crown diameters of *H. erectus* s.l. (HE), *H. antecessor* (HA), European Middle Pleistocene *Homo* (EMPH), Asian Middle Pleistocene *Homo* (AMPH), Neanderthals (NEA), Upper Palaeolithic *H. sapiens* (UPMH), and Holocene humans (HH) (see Supplementary Tables 9, 10). Adjusted Z-score analyses[109,110] were performed on the mesiodistal and buccolingual diameters of TNH2-1 in comparison with the comparative specimens/samples. This statistical method allows the comparison of unbalanced and small samples, which is often the case when dealing with the fossil record, using the Student's t inverse distribution following the formula: $[(x-m)/(s*sqrt(1 + 1/n))]/(Student.t.inverse(0.05;n-1))$, where x is the value of the variable; m is the mean of the same variable for a comparative sample; n is the size of the comparative sample; and s is the standard deviation of the comparative sample (Supplementary Table 11).

## 3D crown tissue proportions

Three variables describing tooth crown tissue proportions were digitally measured on TNH2-1: the percent of the crown volume that is dentine and pulp (Vcdp/Vc; %), the average enamel thickness (3D AET; mm) and the scale-free 3D relative enamel thickness (3D RET)[111–113]. Intra- and interobserver accuracy tests of the measures run by two observers provided differences <5%. Adjusted Z-score analyses[109,110] were performed on the tooth crown tissue proportion parameters for TNH2-1 in comparison with the comparative specimens/samples.

## Geometric morphometric analyses of the enamel-dentine junction (EDJ) shape

We used a diffeomorphic surface matching (DSM) approach to analyse the EDJ conformation. This method constructs shape atlases using mathematical currents (functions to analyse manifolds), where topology of the template is preserved and deformation parameters are optimised independently of the shape parameters[114–116]. We used the Deformetrica v. 4.3 software (https://www.deformetrica.org) to generate a global mean shape with a set of diffeomorphisms relating the global mean shape to each individual and the output (control points and deformation momenta) to perform the statistical analyses, and explore the EDJ shape variation and classify the data. This approach has already been used to study the EDJ shape of fossil hominins and proved to be a reliable method to discriminate between taxa[117,118]. Fitting deformation-based models were applied to the complete EDJ decimated to 50,000 polygons and aligned by Procrustes superimposition. Smooth and invertible deformations combined with a metric of currents that defines a distance between EDJ surfaces allowed us to quantify the degree of shape variation between the investigated specimens. The global mean shape and the deformations functions were then used to classify the data. We also compared the results of the DSM analyses with those obtained with a classic geometric morphometric method based on landmarks and semi-landmarks placed along the EDJ occlusal and cervical margins. More precisely, four anatomical landmarks were placed on the tip of the dentine horn of four primary cusps (protoconid, metaconid, entoconid, and hypoconid) and a first set of semi-landmarks was positioned along the top of the marginal ridge connecting the dentine horns (beginning at the top of the protoconid and moving in a lingual direction), while a second set of semi-landmarks was placed along the cervix starting below the protoconid dentine horn (specifically, the mesio-buccal corner of the cervix) and moving in a mesial direction[39]. Where small parts of the cervix were missing, its location was estimated. In most cases the cervix landmarks were placed on a surface model of the whole dentine crown as this is most appropriate and facilitates placing landmarks on specimens in which missing enamel impacts the completeness of an EDJ surface. Geometrically corresponding semilandmarks[119] were derived in Mathematica (Wolfram Inc.) by using a software routine[120,121] following the measurement protocol detailed in Skinner et al.[122,123]. A smooth curve was interpolated using a cubic spline function. The cubic spline was fitted by starting at the first landmark and moving in a lingual direction to the last point. The interpolated curves were then resampled to achieve identical point counts among specimens. The semi-landmarks of the EDJ marginal ridge were then projected onto the curve, dividing the curve into four sections (protoconid to metaconid: 12 landmarks; metaconid to entoconid: 12 landmarks; entoconid to hypoconid: 24 landmarks; hypoconid to protoconid: 12 landmarks). For the semi-landmarks at the cervix, 30 equally spaced landmarks were generated along the cervix curve.

For both geometric morphometric approaches, we then performed generalised Procrustes analyses, followed by principal component analyses (PCA). We investigated the structure and group separation along the PC scores, checking that the groups already pre-exist in the PCA (Supplementary Fig. 7). Both DSM and landmark-based approaches show a substantial degree of separation between taxa along the first three PC axes (Supplementary Fig. 7). We computed canonical variate analyses (CVA) and between-group principal component analyses (bgPCA). The following three groups were used in both CVA and bgPCA: *H. erectus* s.l. (HE), European Middle Pleistocene *Homo* and Neanderthals (EMPH-NEA), Pleistocene and Holocene *H. sapiens* (MH). TNH2-1 and the two molars (M1 and M2) of the Xiahe mandible were then projected *a posteriori* in the CVA and bgPCA. The CVA was conducted based on the first few principal components explaining ~90% of the total variance[124] (18 in the landmark-based and 24 in the deformation-based analyses; Fig. 5a, Supplementary Fig. 7). We also performed cross-validated canonical variates analyses (CV CVA) still based on the first few principal components explaining ~90% of the total variance for both DSM and landmark-based data (Supplementary Fig. 7). The CV CVA was done first to check the probability that the TNH2-1 specimen represents an M1 or an M2 (Supplementary Table 16) and then with the same groups for the bgPCA to assess the taxonomic prediction (Supplementary Table 17). The bgPCA was computed based on the deformation moments (Fig. 5b) and Procrustes residuals (Supplementary Fig. 7). The DSM and landmark analyses gave similar results for the both CVA and bgPCA, although the DSM method better discriminates the groups (Fig. 5b, Supplementary Fig. 7). The fact that two different GM approaches (DSM and landmarks) reproduce comparable group discrimination and distribution when using either CVA or bgPCA, that are computed with different input data (a selected set of PC scores and the moments/Procrustes residuals, respectively), suggests that the results of the geometric morphometric analyses are robust. We also followed the recommendations of Cardini and Polly regarding the validity of bgPCA[125] by computing the cross-validated bgPCA (CV bgPCA; Supplementary Fig. 7) and

comparing it with the bgPCA (Fig. 5b, Supplementary Fig. 7h), showing that both display the same degree of group discrimination. Results of the CV CVA (Supplementary Table 17) and CV bgPCA show a high level of correct classification (HE: 94.1%; EMPH-NEA: 98.0%; MH: 88.2% for the bgPCA based on DSM with three groups). In order to test further that groups separation in not artificial, we also computed the bgPCA with five groups instead of three (including *H. erectus*, and separating Middle Pleistocene and Late Pleistocene Neanderthals, as well as Pleistocene modern humans and Holocene humans; Supplementary Fig. 7). These analyses show nearly identical taxonomic distinctions than in the bgPCA based on three groups (*H. erectus*, Neanderthals and modern humans; Fig. 5b, Supplementary Fig. 7), indicating that groups separation reflects the biological reality. Altogether, the PCA and bgPCA, CV bgPCA and CVA analyses based on landmarks and DSM are consistent with each other and confirm that the groups observed in the bgPCA are not spurious[126]. The analyses were performed in the packages ade4 v.1.7-6[127] and Morpho v. 2.7[128] for R v.3.4[129]. Allometry was tested in the semi-landmark analyses using multiple regression[130] in which the explanatory variable is the centroid size and the dependent variables are the PC and bgPC scores. In all PCA and bgPCA, the first two components show no allometry (*p*-value > 0.05). The differences between specimens thus represent shape-variation.

**Ancient protein analysis**. Enamel protein extraction followed protocols detailed elsewhere[34]. In short, small enamel samples were demineralised in either hydrochloric acid (HCl) or trifluoroacetic acid (TFA), cleaned up on C18 stagetips and subsequently analysed using nanoLC-MS/MS without protease digestion or cysteine reduction and alkylation.

The proteome of the TNH2-1 enamel was analysed with a previously published workflow[34]. The sample was processed in a clean lab suitable for ancient DNA and protein analysis. The enamel was mechanically cleaned, and two small samples (12 and 13 mg) were removed and powdered for protein extraction. For demineralisation, the enamel powders were suspended in either 1.5 M (5.5% w/w) HCl or 0.88 M (10% w/w) TFA and incubated at room temperature overnight. Peptides were extracted from the supernatant and desalted by solid-phase-extraction using C18 Stagetips[131]. Laboratory blanks were prepared using the same procedure. The peptides were vacuum-dried, reconstituted in 0.1% TFA 5% ACN and quantified using a Nanodrop spectrophotometer at 280 nm. Approximately 500 ng peptides were loaded on a 75 µm inner diameter, 15 cm length column home-packed with 1.9 µm Reprosil-Pur C18 beads (Dr. Maisch) with a laser-pulled silica emitter. The analytes were separated using an EASY-nLC 1200 (Thermo Scientific) and two different gradients using 0.1% TFA 99.9% water as solvent A and 0.1% TFA 80% ACN 19.9% water as solvent B at a flow rate of 350 nL/min. The shorter 45 min gradient increased from 8 to 15% B over 25 min, to 30% B over 15 min, and to 45% B over 5 min. The longer 90 min gradient increased from 8 to 20% B over 75 min, from 20 to 30% B over 10 min and from 30 to 45% B over 5 min. The LC was interfaced with an Orbitrap Q-Exactive HF-X (Thermo Scientific)[132], acquiring full MS scans from 350 to 1400 m/z at 60,000 resolution at 200 m/z, 3e6 AGC target, and 45 ms max. Injection time (IT) was followed by 10 data-dependent MS2 scans with 1.3 m/z isolation width and HCD fragmentation at a normalised collision energy of 28 and measured at 45,000 resolution, 1e5 AGC target, 45 ms max. IT, and a first mass of 100 m/z. The raw files were analysed with Maxquant version 1.6.0.17[133] searching against an in-house curated database containing protein sequences expected in the Hominidae enamel proteome[35,36]. Candidate peptides were generated by "unspecific" digestion and allowed for up to 3 of the following variable modifications: deamidation (NQ), hydroxyproline (P), phosphorylation (STY), oxidation (MW), dioxidation (W), kynurenine formation (W), and arginine-to-ornithine conversion (R), which were chosen based on a previously published open-search strategy[134] using the "dependent peptides" search feature of Maxquant. The Maxquant search also included the approximately 3-centuries old enamel proteome from a human (Ø1952, male, *Homo sapiens*) that was previously prepared and analysed in an identical manner[35], as well as numerous extraction and mass spectrometry blanks.

Extraction and mass spectrometry blanks contained no proteins, indicating clean extraction and instrument conditions during analyses. Subsequently, proteins represented by a single peptide and likely contaminants were removed from analyses. This left a small number of known enamel proteins for further phylogenetic and diagenetic analyses (Supplementary Table 7). Spectra overlapping amino acid positions with sequence differences between Neanderthals, Denisovans, the Xiahe Denisovan, and *H. sapiens* were inspected manually to determine a phylogenetic assignment. Here, spectra with incomplete fragmentation around sites of interest were excluded from consideration. No high-confidence spectra were identified, making it impossible to assign TNH2-1 to any of these three groups based on palaeoproteomic data. A similar inspection of phylogenetically informative sites among great apes indicates that TNH2-1 represents a hominin (Supplementary Fig. 14; Supplementary Table 8). Protein modifications that were previously identified in early Pleistocene enamel[34] were quantified by relative spectral counting. Therefore, the number of peptide-spectrum matches (PSMs) containing a certain PTM was divided by the total number of PSMs containing the respective amino acid(s) that can be modified. The sequence motif analysis of peptides containing a phosphorylated serine residue was done using the R-package "ggseqlogo"[135]. The R scripts for the PTM analyses and construction of Supplementary Fig. 4 are available in the

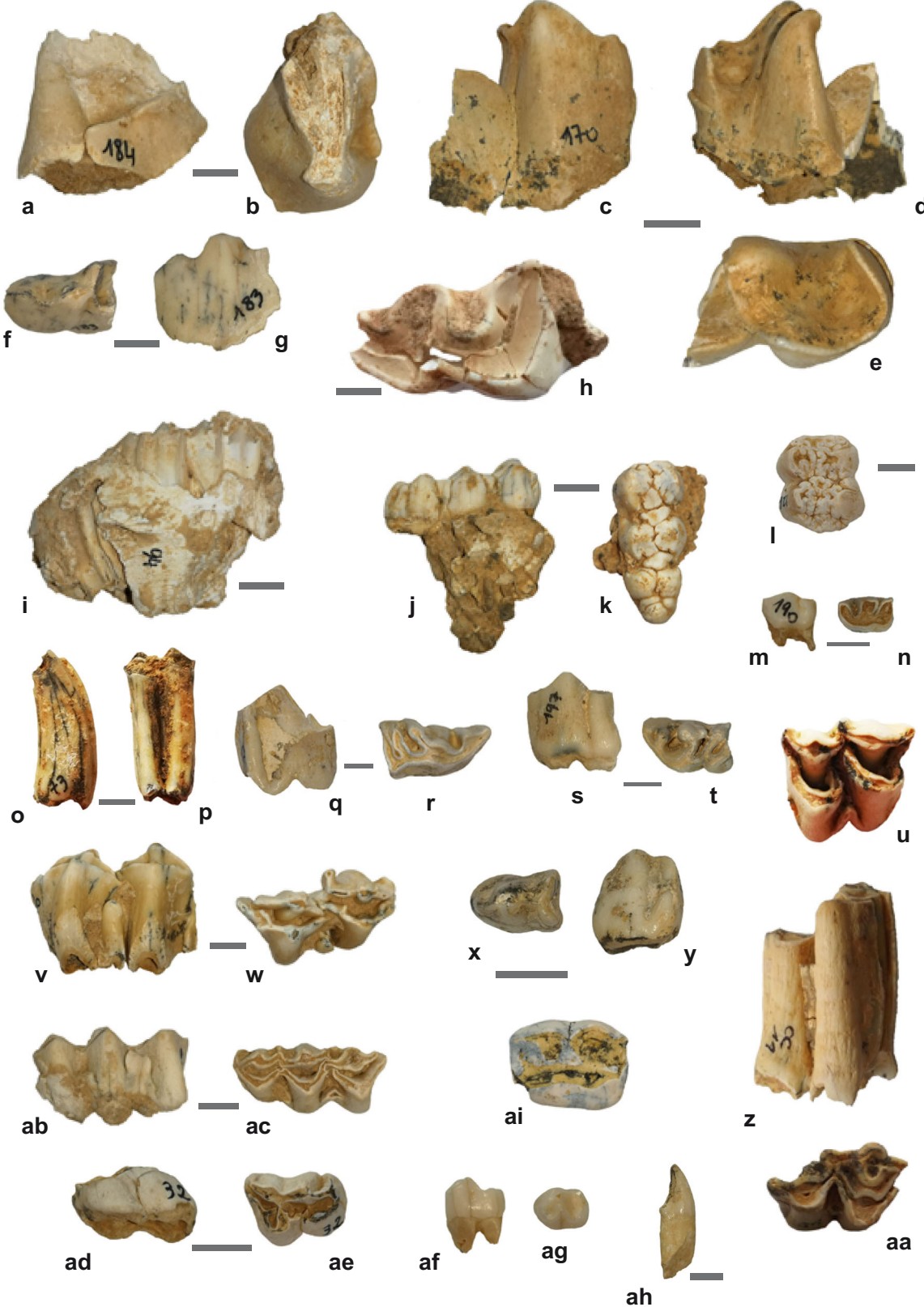

supplementary data 1. Peptide fragment spectra covering sites relevant for phylogenetic placement were annotated using the "Interactive Peptide Spectral Annotator" webtool[136] (Supplementary Fig. 14; Supplementary Data 1) and calibrated peak lists from Maxquant. Raw data and peptide identifications resulting from our MaxQuant analyses are available on ProteomeXchange[137] (PXD018721).

## Associated fauna from Tam Ngu Hao 2 cave

*Description.* Thirty-eight identified teeth (one deciduous and 37 permanent) of suids and numerous small fragments display the morphological pattern of *Sus* with bunodont cusps and cuspids on molars and numerous accessory tubercles. Only 12 teeth (most premolars) out of 38 are complete, with the majority being fragmentary

**Fig. 6 Faunal remains from Tam Ngu Hao 2 cave with some of the most well-preserved teeth. a–b** TNH2-184, fragment of a right D4 of *Dicerorhinus* sp. in anterior (**a**) and occlusal (**b**) views. **c–e** TNH2-170, fragment of a right d4 (germ) of *Rhinoceros* sp. in buccal (**c**), lingual (**d**) and occlusal (**e**) views. **f–g** TNH2-183, fragment of a left d2 of *Rhinoceros sondaicus* in occlusal (**f**) and buccal (**g**) views. **h** TNH2-177, fragmentary left m1/m2 of *Rhinoceros* sp. in occlusal view. **i** TNH2-94, fragmentary left maxillae of an unidentified caprine in buccal view. **j–k** TNH2-117, right m3 of *Sus scrofa* in buccal (**j**) and occlusal (**k**) views. **l** TNH2-126, right M2 of *Sus scrofa* in occlusal view. **m–n** TNH2-190, left p3/p4 of a small-sized cervid in buccal (**m**) and occlusal (**n**) views. **o–p** TNH2-73, left M1/M2 of a caprine in distal (**o**) and buccal (**p**) views. **q–r** TNH2-79, right p3 of *Bos* sp. in buccal (**q**) and occlusal (**r**) views. **s–t** TNH2-197, left p4 of a large-sized cervid in buccal (**s**) and occlusal (**t**) views. **u** TNH2-41, upper left molar of a large-sized cervid in occlusal view. **v–w** TNH2-46, right d4 of *Bos* sp. in buccal (**v**) and occlusal (**w**) views. **x–y** TNH2-61, left p2 of *Bos* sp. in occlusal (**x**) and buccal (**y**) views. **z–aa** TNH2-11, fragment of a right m3 of *Bos* sp. in buccal (**z**) and occlusal (**aa**) views. **ab–ac** TNH2-196, left d4 of a medium-sized cervid in buccal (**ab**) and occlusal (**ac**) views. **ad–ae** TNH2-32, right D2 of a medium-sized cervid in lingual (**ad**) and occlusal (**ae**) views. **af–ag** TNH2-167, right p4 of *Macaca* sp. in buccal (**af**) and occlusal (**ag**) views. **ah** TNH2-152, left lower canine of a female *Macaca* sp. in lateral view. **ai** TNH2-155, left M1 of *Ursus thibetanus* in occlusal view. Scale: 1 cm.

or broken. We did not see significant differences on teeth that would suggest the presence of several species. On two well-preserved m3s (particularly the complete TNH2-117) (Fig. 6j–k), the anterior cingulum is marked and the median accessory cuspids are well developed. It is also notable on one M3 (TNH2-125), with well-developed cingulum and median cusp (hyperconule) behind the first row like in *Sus scrofa* (*versus Sus barbatus*[138]). The dimensions taken on few teeth fall in ranges of *Sus scrofa* from the late Middle Pleistocene to Late Pleistocene Indochinese sites[28,139–141], unlike *Sus lydekkeri* which was much larger[25] (Supplementary Data 2).

Thirty-five permanent teeth of large bovids are massive with a relatively thick enamel (a feature that helps distinguish bovids from cervids of comparable size and among large-sized bovids, *Bos* from *Bubalus*, based on our own observations). Most of the teeth are fragmentary, and within the set of permanent teeth only three (one right p3, one left p2, and one lower right m3) provide systematics input. The p2 is small and massive like in *Bos* as opposed to elongated as in *Bubalus* (Fig. 6x, y). The valleys are also shallow like in *Bos*. The parastylid and the paraconid of the p3 are less developed in comparison with the posterior cuspids. The metaconid is oriented posteriorly. Dimensions of p2s and p3s fall within the size ranges of the modern *Bos gaurus* (Fig. 27 from ref. [142]). On m3, we note a salient entostylid, which forms a deep notch with the hypoconulid. Even if morphological and morphometrical data are not directly comparable, the outline and the orientation of the hypoconulid on the m3 is closer to those of the Late Pleistocene *Bos gaurus frontalis* (Fig. S4c from ref. [143]) (different nomenclatures are available for this taxon: *Bos gaurus*[144]; *Bos frontalis*[145–147]). Five fragmentary milk teeth and one complete left d3 are likely to be associated with this set of permanent teeth. The large bovid of TNH2 cave can be assigned to *Bos* sp., possibly *B.* cf. *frontalis*.

Bovids from TNH2 cave also include one small-sized caprine with three almost complete and three fragmentary isolated teeth. One fragmentary maxilla still covered by hard breccia sediment displays the left side with D4, M1, and M2 still in the bone (TNH2-94) (Fig. 6i). Another fragment preserves the right side with M2 as a germ (TNH2-95). All this material displays the archetypical pattern of the subfamily with rather smooth enamel (not so ridulated as in cervids) and styles and stylids that are more prominent than main cusps and cuspids. One lower molar shows an ectostylid not common in caprines to our knowledge, and observed on one specimen of the modern *Naemorhedus goral* from India (MNHN-specimen 1939–243). Dimensions of the three molars are compatible with the size ranges of a small-sized *Naemorhedus* species (Fig. 34 from ref. [142]), the infraspecific identification of which remains impossible with the available remains (Supplementary Data 2).

Twenty-three teeth (19 permanent and 4 deciduous) exhibit a morphological pattern and dimensions similar to those of a large deer: ridulated enamel, large cusps and cuspids on molars, well-developed ectostylid and entostyle on lower and upper molars, respectively (Fig. 6u). Four well-preserved lower premolars, one fragmentary p2, two p3s and one p4 could belong to *Rusa unicolor*. The p4 is not fully molarized and shows a partial closure of the valley (formed by the metaconid and the paraconid) (Fig. 6s–t). Overall, dimensions of the teeth (Supplementary Data 2) are small and closer to those of the large cervid recovered at Coc Muoi (late Middle Pleistocene, Vietnam[28]) and to those of the Khok Sung site (Middle Pleistocene, Thailand[142]) rather than to those of the Tam Hay Marklot site (Late Pleistocene, Laos[141]).

A set of identifiable small teeth of Artiodactyla, most fragmentary, suggests the presence in the assemblage of small and medium-sized cervids (Fig. 6m–n, ab–ae). Among twenty-four teeth, one m3 and one m1/m2, one elongated lower p2 (parastylid straight, metaconid oriented posteriorly), and two p3/p4 all worn, one elongated d2 and two D4s, along with several P3/P4s have a morphology and dimensions consistent with those of *Muntiacus* (Supplementary Data 2). The identification at the species level, however, is impossible with the available material. Six teeth most likely belong to a medium-sized cervid, among which two worn and fragmentary molars, one incisor, one d4 (TNH2-196), one incomplete milk tooth, and one D2 (TNH2-32). It is morphologically and metrically close to the small-sized cervid from Tam Hay Marklot, tentatively identified as *Rucervus eldii* (Late Pleistocene, Laos[141]), and to that of Coc Muoi (late Middle Pleistocene, Vietnam[28]).

The rhinocerotid sample mostly includes fragments of isolated teeth. Nine permanent teeth (P/p and M/m) and three milk teeth are too fragmentary to allow for more precise identification than *Rhinoceros* sp. (TNH2-170, TNH2-177, and

TNH2-179) or Rhinocerotina indet. (i.e. *Rhinoceros* or *Dicerorhinus*) (Fig. 6c–e, h). Nevertheless, morphology and dimensions of nine other specimens are more informative. Among them, two incomplete teeth have morphological features and dimensions fully compatible with *Rhinoceros sondaicus*: i) a left p3 (TNH2-188) with a short, thick and smooth labial cingulid and a shallow ectolophid groove, perfectly matching that of the jaw n°305c from Soember Waroe (Middle Pleistocene, Java; Fig. 13 from ref. [148]) and the p3 CM1027 from Coc Muoi (late Middle Pleistocene, Vietnam[28]); and ii) a left d2 (TNH183) which is narrow with a bulbous paralophid and no anterolabial groove that falls in the range of the large *R. sondaicus* sample from Coc Muoi[28]. A small and fragmentary right D4 (TNH2-184) with a sigmoid protoloph and an anteriorly-constricted protocone documents *Dicerorhinus* sp., but its fragmentary condition precludes any assignment at the species level. One tapir (*Tapirus* sp.) completes perissodactyl taxonomic diversity at TNH2 cave, with the posterior lophid (hypolophid) of a fragmentary d4 (TNH2-178) that is too small to be assigned to *Megatapirus* (width: 19.5 mm instead of 22–23 mm in the latter taxon) and is instead compatible with both *T. indicus indicus* and *T. i. intermedius*[25]. Several enamel fragments, both thick and wrinkled, unambiguously point to the presence of a stegodon elephantiform (*Stegodon* sp.).

Small-sized Carnivora are represented by two premolars, one fragmentary (TNH2-9) and one complete (TNH2-22), but their identification is impossible, even at the genus level. One lower canine could belong to a small-sized felid. A distal part (talonid) of one lower p4 (TNH2-165) is characteristic of Viverridae, possibly *Paradoxurus* sp.

Large Carnivora are documented by few teeth, among which two canines (TNH2-159 and 182) are fragmentary (Carnivora indet.). One fragmentary m3 and one worn M1 (TNH2-155) are referred to *Ursus thibetanus* (Fig. 6ai). The M1 size is close to that of *Ursus thibetanus kokeni* from Yenchingkuo (Middle Pleistocene, China[25]) and slightly larger than the M1s from Duoi U'Oi (Late Pleistocene, Vietnam[139]) or Coc Muoi (late Middle Pleistocene, Vietnam[28]). However, this comparison is based on small data sets (less than three specimens in each site). Although incomplete, two teeth can be assigned to *Ailuropoda* sp. (TNH2-154 and 152). The assignment of another small fragment to this genus is questionable (TNH2-153).

Ten teeth belong to a cercopithecid (*Macaca* sp.). Dimensions taken on the most complete teeth (Supplementary Data 2), one p4 (TNH2-167) (Fig. 6af–ag), one M3 (TNH2-162), one M1/M2 (TNH2-205) and one m3 (TNH2-163), fall within ranges of the two middle-sized species *M. mulatta* and *M. nemestrina* (Fig. 4 from ref. [149]) on M3 and m3; personal data on p4). The well-preserved lower p4 (TNH2-167) presents a small anterior basin oriented lingually and a large and rounded posterior basin as in *M. nemestrina*. Due to the small data set, we cautiously assigned the specimens to *Macaca* cf. *nemestrina*. One fragmentary molar also testifies to the presence of an orangutan (*Pongo* sp.) at the Tam Ngu Hao site.

Eighteen teeth, most worn, belong to a large-sized porcupine (*Hystrix* sp.) (Supplementary Data 2). Length of two right p4s, two right m1/m2 and four upper molars are on average greater than those of the extant Asian porcupine *Hystrix brachyura*, and one m1/m2 (TNH2-143) is even larger than those of the other extant species *Hystrix indica* (Fig. 4 from ref. [150]). Overall, dental dimensions fall within the range of Pleistocene *Hystrix* from several Chinese and Indochinese localities. An assignment to the relatively large-sized species *H. kiangsenensis* cannot be confirmed.

**Comments on associated fauna from Tam Ngu Hao 2 Cave.** Despite the poor preservation of recovered specimens, the Tam Ngu Hao 2 cave assemblage is largely similar to those known in southern China and northern Indochina and, to a lesser extent, Java in the Middle Pleistocene[25–27,151,152]. The estimated age range of the fauna (151 +/− 37 kyr) based on the combined US-ESR dating of three teeth fully agrees with the biochronological context given by the faunal assemblage.

**Reporting summary**. Further information on research design is available in the Nature Research Reporting Summary linked to this article.

## Data availability
The mass spectrometry proteomics data generated in this study have been deposited in the in the ProteomeXchange Consortium via the PRIDE partner repository with the

dataset identifier PXD018721, including the used proteomic reference database. The hominins teeth and faunal teeth measurements data generated in this study are provided in the Supplementary Data 2 file. The script for Bayesian modelling the Tam Ngu Hao 2 age estimates is provided in Supplementary Fig. 13. TNH2-1 tooth and all the faunal teeth TNH2-2 to TNH2-208 are housed at the National Museum under the responsibility of the Ministry of Information and Culture of Lao PDR. A surface scan of THN2-1 tooth and its EDJ is publicly available in the Human Record archive (https://human-fossil-record.org), register an account and afterwards click the Museum of Lao PDR link on the left hand side of the page). Source data are provided with this paper.

## Code availability

The code used for Bayesian modelling the Tam Ngu Hao 2 age estimates is provided as a Supplementary Fig. 13.

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

## Acknowledgements

The team dedicates this work to Mr Thongsa Sayavongkhamdy. We thank the Ministry of Information, Culture and Tourism of Lao PDR for encouraging and supporting our work, marking almost 20 years of collaboration. We thank the authorities of Xon district, Hua Pan Province and the villagers of Long Gua Pa village for their continuous support of our numerous years of fieldwork. We also thank many curators and colleagues who granted access to the comparative fossil and recent hominin specimens for scanning, as well as the online sharing platforms of the Nespos society (http://www.nespos.org) and ESRF Paleontological database (http://paleo.esrf.eu). We acknowledge A. Mazurier and R. Macchiarelli (University of Poitiers), A. Bravin, C. Nemoz and P. Tafforeau (ESRF Synchrotron), P. Bayle and F. Santos (University of Bordeaux), O. Kullmer and F. Schrenk (Senckenberg Museum), F. Bernardini and C. Tuniz (ICTP Trieste), J. Braga and J. Dumoncel (University of Toulouse) for analytical support, microtomographic scanning and sharing of material. We gratefully acknowledge support from the CNRS/IN2P3 Computing Center (Lyon - France) for providing computing and data-processing resources needed for this work. Our thanks also go to Christine Lefèvre, Joséphine Lesur and Aurélie Verguin (Laboratoire Mammifères et Oiseaux, Anatomie comparée, MNHN, Paris) for access to comparative mammalian collections. Warm thanks to Jeremy Kazan (Taxidermie, MNHN, Paris) for having, as usual, wonderfully casted and reproduced the specimen. M.W.M. thanks George Morgan and Mark Keene at Adelaide Petrographics for expertly making the micromorphology thin sections. This research has been funded by the Australian Research Council (ARC) Future Fellowship award (FT180100309) (M.W.M.), the Australian Research Council (ARC) Discovery grant (DP170101597) (K.E.W.), and the Australian Research Council (ARC) LEF Grant (LE200100022) (R.J.-B.), the Marie Skłodowska Curie Individual Fellowship (no. 795569) (F.W.). Marie Skłodowska-Curie European Training Network (ETN) TEMPERA, a project funded by the European Union's Framework Program for Research and Innovation Horizon 2020 (no. 722606) (E.C., J.V.O., P.R.), the European Research Council (ERC) under the European Union's Horizon 2020 research and innovation programme (no. 948365) (F.W.), the VILLUM FONDEN (no. 17649) (E.C.). European Union's Framework Program for Research and Innovation Horizon 2020 (no. 819960) (M.M.S.), the Novo Nordisk Foundation (grant number NNF14CC0001) (P.L.R.), the National Geographic Society (NGS-399R-18) (L.S.), the MNHN/Université Paris Diderot/Sorbonne Paris Cité, France (F.D.), the CNRS/Université de Paris, France; Université de Strasbourg, France (A.M.B.). Department of Human Evolution, Max Planck Institute for Evolutionary Anthropology, Leipzig, Germany (A.M.B.) and the Sodipram company (La Compagnie du Lit), France (F.D.).

## Author contributions

Conceptualisation: F.D., L.S., A.M.B., T.S., V.S., T.L. Methodology: C.Z., M.M.S., K.E.W., R.J.-B., R.M.B. Formal analysis: C.Z., M.M.S., A.M.B., P.O.A., M.W.M., L.P., X.S., J.-J.H., P.D., J.-L.P., Q.B., F.W., P.L.R., J.V.O., E.C., H.M.C., K.E.W., R.J.-B., J.-X.Z., R.M.B. Investigation: F.D., C.Z., L.S., P.D., J.-L.P., E.S., S.F., T.D., Q.B., A.M.B., F.W., P.L.R., J.V.O., E.C., H.M.C., K.E.W., R.J.-B., J.-X.Z., R.M.B. Writing – original draft: F.D., C.Z., L.S. Writing – review & editing: F.W., P.D., A.M.B., P.O.A., M.W.M., K.W., C.G., L.V., M.S., S.B., P.S., D.S., E.P.E., F.A., F.C., N.B., A.Z., J.-J.H., E.W. Supervision: F.D., L.S., A.M.B. Project administration: F.D., L.S., A.M.B. Funding acquisition: F.D., L.S., A.M.B., K.E.W., R.J.-B., E.C., F.W., J.-J.H., E.W.

## Competing interests

The authors declare no competing interests.

## Additional information

[1]Lundbeck Foundation GeoGenetics Centre, Globe Institute, University of Copenhagen, Copenhagen, Denmark. [2]Eco-anthropologie (EA), Muséum national d'Histoire naturelle, CNRS, Université de Paris, Musée de l'Homme 17 place du Trocadéro, 75016 Paris, France. [3]Univ. Bordeaux, CNRS, MCC, PACEA, UMR 5199, 33600 Pessac, France. [4]'Traps' Luminescence Dating Facility, School of Natural Sciences, Macquarie University, Sydney, NSW, Australia. [5]Geoarchaeology and Archaeometry Research Group (GARG), Southern Cross University, Lismore, NSW, Australia. [6]Centre for Anthropological Research, University of Johannesburg, Johannesburg, Gauteng Province, South Africa. [7]Ecole et Observatoire des Sciences de la Terre, Institut de Physique du Globe de Strasbourg (IPGS), UMR 7516 CNRS, Université de Strasbourg, Strasbourg, France. [8]Archaeology, College of Humanities and Social Sciences, Flinders University, Sturt Road, Bedford Park, Adelaide, SA, Australia. [9]Section for Evolutionary Genomics, Globe Institute, University of Copenhagen, Copenhagen, Denmark. [10]The Novo Nordisk Foundation Center for Protein Research, University of Copenhagen, Copenhagen, Denmark. [11]School of Anthropology and Conservation, University of Kent, Canterbury, UK. [12]Department of Human Evolution, Max Planck Institute for Evolutionary Anthropology, Leipzig, Germany. [13]Department of Medical Education, Creighton University School of Medicine, Omaha, NE, USA. [14]Université de Strasbourg, Laboratoire Image, Ville Environnement, UMR 7362, UdS CNRS, Strasbourg, France. [15]Spitteurs Pan, Technical Cave Supervision and Exploration, La Chapelle en Vercors, France. [16]Institut des Sciences de l'Évolution, Univ Montpellier, CNRS, IRD, Montpellier, France. [17]Key Laboratory of Vertebrate Evolution and Human Origins, Institute of Vertebrate Paleontology and Paleoanthropology CAS, Beijing, China. [18]State Key Laboratory of Palaeobiology and Stratigraphy, Nanjing Institute of Geology and Palaeontology CAS, Nanjing, China. [19]CAS Center for Excellence in Life and Paleoenvironment, Beijing, China. [20]School of Earth and Environmental Sciences, University of Queensland, Brisbane, QLD, Australia. [21]School of Geography and the Environment, University of Oxford, Oxford, UK. [22]Ministry of Information, Culture and Tourism, Vientiane, PDR, Laos. [23]Museum d'histoire Naturelle de La Rochelle, La Rochelle, France. [24]IRD, DIADE, Montpellier, France. [25]Department of Biomedical Sciences, University of Minnesota Medical School Duluth, Duluth, MN, USA. [26]Université de Paris, BABEL CNRS UMR, 8045 Paris, France. [27]Collège de France, Paris, France. [28]Department of Zoology, University of Cambridge, Cambridge, UK. [29]Wellcome Trust Sanger Institute, Cambridge, UK. [30]Danish Institute for Advanced Study, University of Southern Denmark, Copenhagen, Denmark. [31]Department of Anthropology, University of Illinois at Urbana-Champaign, Urbana, IL, USA. [32]Carle Illinois College of Medicine, University of Illinois at Urbana-Champaign, Urbana, IL, USA. [33]These authors contributed equally: Fabrice Demeter, Clément Zanolli, Laura Shackelford. [34]Deceased: Thongsa Sayavongkhamdy. ✉email: f.demeter@sund.ku.dk; clement.zanolli@gmail.com; llshacke@illinois.edu

