## [Peer Review File · Nature Communications]

A Middle Pleistocene Denisovan molar from the Annamite Chain of northern LaosREVIEWER COMMENTS

Reviewer #1 (Remarks to the Author):

In paleoanthropology, Southeast Asia is poorly known compared to other regions of the Old World. Given that fact, any hominin fossil that is found in the region is worth further investigation. In this regard, the authors present a new hominin fossil and conducted just about every type of analysis that could be done to determine its taxonomic assignment. Although the currently very popular paleoproteomics analysis failed to yield robust results, the more standard metric and geometric morphometric analysis of the tooth suggested it could be assigned to either Neanderthals or Denisovans. If this fossil can be tied to the better known Maba cranium from southern China, a stronger argument for possible Neanderthals in the region might be possible. This paper should eventually be published pending revisions. More specific comments are below.

Comments:

Page 3, first line. The Wu and Athreya citation is fine here, but there are many citations that should be added given the complexity of this record and how much has been discussed over the past several decades by more people than just Wu and Athreya. To name but a few, I would add Pope 1992, Bae, 2010, Xing et al., 2015; and even the recent paper by Roksandic et al (In Press) that also observes that the Middle Pleistocene hominin fossil record in Asia is quite different from Europe and Africa.

Page 3, starting line 3. The Harbin fossil most closely aligns with Dali. Whether one wants to group the rest of these aforementioned fossils into *Homo longi* is a matter of some strong debate in Chinese paleoanthropology (not just Chinese researchers, but more inclusively, those who work with Chinese fossils). I would guess even the Chinese coauthors here may have some reservations about this as well. Recommend to rewrite in a way that is less assertive and more open to alternative ideas.

Page 3, starting line 7. Not sure a paper about *H. heidelbergensis* from 2012 nor an article by a journalist is appropriate here as taking in the full range of the discussion of the current Chinese hominin fossil record. Citing Liu et al., 2014; Pope, 1992; Bae, 2010 to name a few, would be more useful here.

Page 3, last sentence. Observation. A bit surprised that the authors cannot determine whether a completely developed, but unworn molar can be assigned as a first or second molar. I will read further, but I hope the authors include more detailed discussion in the main text as to why they cannot determine this, given the importance of its identification as either an M1 or an M2 in many metric and geometric morphometric studies of teeth (as surely some of the coauthors here who are well versed in this area realize).

Page 4, Dating section. It might be worth it to acknowledge the complexity of these karst cave systems and how difficult it is to determine reliable ages based on U-series from associated fossils. For example, Michel et al. (2016) provide valid criticisms of an earlier study where the cave system, for all intents and purposes, looks like a fluvial transported assemblage. It would be worthwhile to discuss this important point. From reading the text, the context looks different (cf. Bae et al., 2014), but it is an important point to raise. Because of this complexity, readers will likely ask this same question.

Page 8, bottom. This is an interesting suggestion that the tooth may actually represent a Neanderthal. If the authors want to make this argument, they should cite the studies of the Maba cranium (Pope, 1992; Wu and Poirier, 1995; Xiao et al., 2014) that acknowledge that the Maba fossil was long suggested to have been a Neanderthal.

References

Bae, C.J., 2010. The late Middle Pleistocene hominin fossil record of eastern Asia: Synthesis and review. *Yearbook of Physical Anthropology* 53, 75-93.

Bae, C.J., Wang, W., Zhao, Z.X., Huang, S.M., Tian, F., Shen, G.J., 2014. Modern human teeth from Late Pleistocene Luna cave (Guangxi, China). *Quaternary International* 354, 169-183.

Liu, W., Wu, X.J., Xing, S., Zhang, Y.Y., 2014. *Human Fossils in China* (in Chinese). Science Press, Beijing.

Michel, V., Valladas, H., Shen, G.J., Zhao, J.X., Shen, C.C., Valensi, P., Bae, C.J., 2016. Earliest modern *Homo sapiens* in China? *Journal of Human Evolution* 101, 101-104.

Pope, G.G., 1992. Craniofacial evidence for the origin of modern humans in China. *Yearbook of Physical Anthropology* 35, 243–298.

Roksandic, M., Radovic, P., Wu, X.J., Bae, C.J. In Press. Resolving the “Muddle in the Middle”: the case for *Homo bodoensis* sp. nov. *Evolutionary Anthropology*.

Wu, X.Z., Poirier, F.E., 1995. *Human Evolution in China*. Oxford: Oxford University Press.

Xiao, D.F., Bae, C.J., Shen, G.J., Delson, E., Jin, J., Webb, N.M., Qiu, L.C., 2014. Metric and geometric morphometric analysis of new hominin fossils from Maba (Guangdong, China). *Journal of Human Evolution* 74, 1-20.

Xing, S., Martínón-Torres, M., Bermúdez de Castro, J.M., Wu, X.J., Liu, W., 2015. Hominin teeth from the early Late Pleistocene site of Xujiayao, northern China. *American Journal of Physical Anthropology* 156, 224–240.

Reviewer #2 (Remarks to the Author):

Genetic analyses of contemporary humans, showing high amounts of Denisovan gene flow into Melanesian and some SE Asian populations have for a long time indicated that the Denisovans were likely present in SE Asia as well. Up until now the lack of Middle to Late Pleistocene hominin fossil remains from the region has strongly limited our ability to test this hypothesis.

In this paper, Demeter and colleagues describe the discovery of an isolated hominin lower molar from Tam Ngu Hao 2 in Laos. Using U-series, ESR and luminescence dating in a Bayesian model they establish that the specimen dates to the end of the Middle Pleistocene. They then use paleoproteomics, morphometrics and the morphology of the specimen to show that it is similar to Neanderthals and the molars of the likely Denisovan mandible from Xiahe, Tibet (comparisons with the material from Denisova cave are not easily possible, as no lower molars are known from there). This is an important discovery, despite the rather fragmentary fossil, as it is the first specimen that could represent the hypothesized Denisovan population in the region.

In general, I feel that the paper does an admirable job at extracting information from this specimen, and I recommend its acceptance.

Some more detailed comments:

Line 82-95: These sentences are a bit conflicting - the authors first list a number of MP specimens from E Asia, then they say that “they were suggested to belong to an Asian sister taxon of Neanderthals, the Denisovans”, and finally they write that “The small number of fossils currently attributed to this group” listing the specimens from Denisova and Xiahe. I assume the authors mean that only the material from Denisova and Xiahe can be securely attributed to Denisovans, while the other material is only potentially Denisovan, but I think they need to make this clearer. The way it is phrased at the moment is somewhat confusing.

Line 96: Australian aborigines should also be mentioned here, as they have comparable amount of Denisovan introgressed DNA as Papuans

Line 121: “and isotopic analyses” - what kind of isotopic analyses were performed? They do not seem to be reported in the paper.

Line 194-197: “This is in line with previous research, which indicated that closely related hominin populations can be distinguished based on dentine and bone proteomes, while enamel proteomes are less informative in the context of close phylogenetic proximity.” If dentine proteomes would be more informative, why were they not performed? Or were these unsuccessful?

Line 231: “and the upper molar of Denisova 4 (3D RET: 15.27)” Where does this data come from? This specimen is not listed in the SI, and there is no citation for this information either

Line 241 - 255, SI Line 690 and following: The authors used a between group Principal Component Analysis to compare shape variation in the EDJ of the various groups. As pointed out by Bookstein (2019, *Evolutionary Biology*) and by Cardini et al (2019, cited in the paper) between group PCAs present several pitfalls when used in high p/n settings, ie. situations with a high number of variables relative to the number of specimens included. One of the largest problems is that this method can introduce spurious intergroup differences, especially in cases where the sample sizes per group differ strongly, or are low (<10). Both the GMM analyses, the one based on semi landmarks (90 semi landmarks x 3 dimensions - 270 variables), but especially the DSM approach are problematic in this context. Cardini and Polly (2020, cited in the paper) do offer some workarounds (such as cross-validated bgPCA scores), but I am not sure whether these are sufficient, especially for the DSM approach.

I want to emphasize that I do not think that this invalidates the authors' conclusions, but I would caution against overinterpreting these plots. In most of the regular PCAs (Fig S7 a,b & d) TNH2-1 plots closest to Homo erectus, which is quite interesting.

Line 246: “Along CN2” Should this be along CV2?

Line 252: “along CV’ ” Should this be CV1?

Line 273-275: “The similarities with Neanderthals that we observe do not preclude TNH2-1 from belonging to this taxon and would make it the south-eastern-most Neanderthal fossil ever discovered.” I find this sentence confusing. Why would similarities preclude a specimen from belonging to a taxon? Do the authors mean differences?

Supplementary materials

SI Line 609-11: That the metameric variation in Denisovans is different from other groups is also supported by the difficulty in identification of the Denisova 4 and 8 molars as M2s or M3s.

SI Fig S5. : “Denisovan M1”, “Denisovan M2”: It might be better to label these as Xiahe 1, M1, and Xiahe 1, M2.

SI Fig S6: M1s and M2s are differentiated by the fill colour in these box and whisker plots. This does not work for the AMPH group, which is represented by a single specimen (I assume this is the M2 of Xiahe) and thus has no box. It would be the easiest to simply label this as “Xiahe M2”.

SI Fig S9., A-E: There seems to be some error with this image, panels A-E are only visible as a thin strip along the upper part of the page. I assume this happened after submission (and is not the fault of the authors), but was the case every time I downloaded the pdf.

SI Line 970: “Halong cave” Should be Hualong cave

Reviewer #3 (Remarks to the Author):

The paper deals with the discovery of a human tooth found in a Middle Pleistocene breccia block in a cave site of Laos.

I will review only the dating part after some general comments on the paper.

First I would like to mention that no prehistoric context is given in the main text. The interest of the paper would be largely increased by presenting the “pre-hoabinian” lithic artefacts for instance.

I remain skeptical with the stratigraphic sketch presented in the paper. I am not sure that we are in presence of two formations due to different hydrogeologic phases. I would be inclined for a karstic underdrawing of a single formation of which only the finest particules have filled the lower part, the coarsest part being more easily carbonated. The layers are poorly described and it seems that taphonomic considerations are not really taken into account.

Fauna is associated to those of East Asia. The most recent stegodon is dated in China to less than 20 ka should at least be mentioned (reference under).

A. Ma, H. Tang, On the discovery and significance of a Holocene Ailuropoda–Stegodon fauna from Jinhua, Zhejiang, *Vertebr. Palasiat.* 30 (1992) 295–312.

OSL

Did the authors measure Radon (^{222}Rn) by Gamma ray spectrometry ? You mentioned ^{210}Pb . Did you measure it at 46 Kev? If no, the best possibility would be to measure the radon by using ^{214}Pb at 352keV of ^{214}Bi at 609 keV.

I do not see any relationship between radium loss and increase of clasts! Could you be more explicit please ? According to what I measured, Even in the closest systems like some uranium standard, I observed at least 17% radon loss (I talk about radon). The explanation on the difference dose due to K seems more reasonable.

Does the important overdispersion values may suggest a pluri-origin of your minerals? Fig S10 is of very bad quality and it is not possible to read what is written for G,H, I schemes. Thank you to improve for clarity.

U-series

The authors write “A better estimate of the most reliable correction is based on the assumption that the real detrital $^{230}\text{Th}/^{232}\text{Th}$ activity ratio should be lower than the bulk-Earth value of 0.825”.

However, recent analyses in speleothems revealed higher $^{230}\text{Th}/^{232}\text{Th}$ than bulk earth (Carolin et al., 2016; Labonne et al., 2002). So what are the arguments for using such a low ratio in the paper? It would worth to add the fitting of the line described by the $^{234}\text{U}/^{232}\text{Th}$ against $^{230}\text{Th}/^{232}\text{Th}$. Frankly speaking, is it possible to get reliable corrected ages with such high detrital components? In other terms, do the corrected ages have a real meaning in this case?

Carolin, S.A., Cobb, K.M, Lynch-Stieglitz, J., Moerman, J.W., Partin, J.W., Lejau, S., Malang, J., Clark, B., Tuen, A.A., Adkins, J.F. 2016. Northern Borneo stalagmite records reveal West Pacific hydroclimate across MIS 5 and 6. *Earth Planet. Sci. Lett.* 439: 182-193.

Labonne, M., Hillaire-Marcel, C., Ghaleb, B., Goy, J.-538 L. 2002. Multi-isotopic age of dirty speleothem calcite: an example from Altamira Cave, Spain. *Quaternary Science Reviews.* 21:1099-1110.

US-ESR on teeth

The three analyzed teeth come from the same location in LU2. I suggest to recalculate the age without taking into account of any water in the enamel. Secondly, the external dose rate ranges between 1100 and 1200 Gy/a and represents 80 to 93% of the total dose. I wonder how the empty space is taken into account in the external dose determination? In other terms what is the explanation of such an empty space in the sequence? Thank you to mention what % of water was used for the sediment? Does it worth to include the CC10 age in the weighted meaning age? I do not think so! In that case we would better be in a MIS5 infilling!

What is the meaning of Bayesian model without a real stratigraphic sequence? Why did you suppress the LCC3 age?

In conclusion, it seems that the paper lacks of contextualization and that the dating asks for some important issues before accepting it for a publication.

REPLY TO THE REVIEWERS COMMENTS

Dear Reviewers, we thank you for your comments and appreciate your suggestions. We addressed one by one all the questions below and we also modified both main and Supplementary material texts and figures, supplying a great deal of data that demonstrates the context of the fossils and the taphonomic history of the tooth under investigation.

Also, after consultation with the co-authors, we feel that our current title doesn't accurately reflect our conclusions and we would like to suggest a slight modification of the title, introducing the notion of a 'possible Denisovan' instead of the much broader genus Homo. We feel that this is more appropriate given that—as outlined in the discussion—even though we cannot definitively discriminate between the only two possibilities, Denisovan and Neanderthal, the most parsimonious scenario that fits with the current state of knowledge is that the tooth belongs to a Denisovan. Consequently, our current title:

"A Middle Pleistocene Homo from the Annamite Chain of northern Laos"

Would become with your agreement:

"A possible Middle Pleistocene Denisovan from the Annamite Chain of northern Laos"

Reply to the Reviewers

Reviewer #1 (Remarks to the Author):

In paleoanthropology, Southeast Asia is poorly known compared to other regions of the Old World. Given that fact, any hominin fossil that is found in the region is worth further investigation. In this regard, the authors present a new hominin fossil and conducted just about every type of analysis that could be done to determine its taxonomic assignment. Although the currently very popular paleoproteomics analysis failed to yield robust results, the more standard metric and geometric morphometric analysis of the tooth suggested it could be assigned to either Neanderthals or Denisovans.

If this fossil can be tied to the better known Maba cranium from southern China, a stronger argument for possible Neanderthals in the region might be possible.

This paper should eventually be published pending revisions.

Reply: We thank you for your comments and we agree with you. Even if we would like to link the new tooth from Laos to more East Asian specimens (like the Maba, Dali or Harbin skulls), this is not possible at the moment, but the publication of this study could help comparisons with Asian fossils found during the last century and that could be related to either Denisovans or Neanderthals.

More specific comments are below.

Comments:

Page 3, first line. The Wu and Athreya citation is fine here, but there are many citations that should be added given the complexity of this record and how much has been discussed over the past several decades by more people than just Wu and Athreya. To name but a few, I would add Pope 1992, Bae, 2010, Xing et al., 2015; and even the recent paper by Roksandic et al (In Press) that also observes that the Middle Pleistocene hominin fossil record in Asia is quite different from Europe and Africa.

Reply: Thank you for the suggestion. We have added the references by Bae (2010), Xing et al. (2015) and Roksandic et al. (in press).

Page 3, starting line 3. The Harbin fossil most closely aligns with Dali. Whether one wants to group the rest of these aforementioned fossils into *Homo longi* is a matter of some strong debate in Chinese paleoanthropology (not just Chinese researchers, but more inclusively, those who work with Chinese fossils). I would guess even the Chinese coauthors here may have some reservations about this as well. Recommend to rewrite in a way that is less assertive and more open to alternative ideas.

*Reply: We agree and we did not intend to support any particular position, we just reported the recent studies on the Harbin cranium. However, to clarify we have changed the sentence as follows: The recent description and analysis of the Harbin cranium from China reignited the debate by suggesting its attribution to a new species named *Homo longi* 8, but the taxonomic attribution of this specimen is a matter of contention.*

Page 3, starting line 7. Not sure a paper about *H. heidelbergensis* from 2012 nor an article by a journalist is appropriate here as taking in the full range of the discussion of the current Chinese hominin fossil record. Citing Liu et al., 2014; Pope, 1992; Bae, 2010 to name a few, would be more useful here.

Reply: Thanks, we have now removed the reference by Gibbons (2021) and cited Bae (2010) and Liu et al. (2014).

Page 3, last sentence. Observation. A bit surprised that the authors cannot determine whether a completely developed, but unworn molar can be assigned as a first or second molar. I will read further, but I hope the authors include more detailed discussion in the main text as to why they cannot determine this, given the importance of its identification as either an M1 or an M2 in many metric and geometric morphometric studies of teeth (as surely some of the coauthors here who are well versed in this area realize).

Reply: As noted in the Supplementary Materials file, the geometric morphometric analyses that we conducted suggested that the M1 and M2 of the Xiahe mandible that markedly resemble TNH2-1 are not conclusive (while we know their metameric position). This could be due to the fact that Asian Neanderthals or Denisovan molars are not represented in our

comparative sample. According to our statistical analyses, TNH2-1 might more likely represent a M2, but we prefer to be more cautious and regard it as M1/2.

Page 4, Dating section. It might be worth it to acknowledge the complexity of these karst cave systems and how difficult it is to determine reliable ages based on U-series from associated fossils. For example, Michel et al. (2016) provide valid criticisms of an earlier study where the cave system, for all intents and purposes, looks like a fluvial transported assemblage. It would be worthwhile to discuss this important point. From reading the text, the context looks different (cf. Bae et al., 2014), but it is an important point to raise. Because of this complexity, readers will likely ask this same question.

Reply: We thank the Reviewers for their comments and completely agree that sediment deposition and carbonate formation are complex processes in cave environments. In order to pre-empt any questions as to the context of the fossils and the arrangement of the stratigraphy we have paid special attention to these aspects of the study including a detailed geological and micro-stratigraphic description of the sequence (details in supplementary materials). The cramped and awkward conditions in the chamber precluded the capturing of images that fully demonstrate the context of the fossils and the dated material.

Page 8, bottom. This is an interesting suggestion that the tooth may actually represent a Neanderthal. If the authors want to make this argument, they should cite the studies of the Maba cranium (Pope, 1992; Wu and Poirier, 1995; Xiao et al., 2014) that acknowledge that the Maba fossil was long suggested to have been a Neanderthal.

Reply: Thank you for the suggestion. We have now added a comment and cited Wu and Poirier (1995) and Xiao et al. (2014) in the Discussion on page 10.

References

Bae, C.J., 2010. The late Middle Pleistocene hominin fossil record of eastern Asia: Synthesis and review. *Yearbook of Physical Anthropology* 53, 75-93.

Bae, C.J., Wang, W., Zhao, Z.X., Huang, S.M., Tian, F., Shen, G.J., 2014. Modern human teeth from Late Pleistocene Luna cave (Guangxi, China). *Quaternary International* 354, 169-183.

Liu, W., Wu, X.J., Xing, S., Zhang, Y.Y., 2014. *Human Fossils in China* (in Chinese). Science Press, Beijing.

Michel, V., Valladas, H., Shen, G.J., Zhao, J.X., Shen, C.C., Valensi, P., Bae, C.J., 2016. Earliest modern *Homo sapiens* in China? *Journal of Human Evolution* 101, 101-104.

Pope, G.G., 1992. Craniofacial evidence for the origin of modern humans in China. *Yearbook of Physical Anthropology* 35, 243–298.

Roksandic, M., Radovic, P., Wu, X.J., Bae, C.J. In Press. Resolving the “Muddle in the Middle”: the case for *Homo bodoensis* sp. nov. *Evolutionary Anthropology*.

Wu, X.Z., Poirier, F.E., 1995. Human Evolution in China. Oxford: Oxford University Press.

Xiao, D.F., Bae, C.J., Shen, G.J., Delson, E., Jin, J., Webb, N.M., Qiu, L.C., 2014. Metric and geometric morphometric analysis of new hominin fossils from Maba (Guangdong, China). *Journal of Human Evolution* 74, 1-20.

Xing, S., Martín-Torres, M., Bermúdez de Castro, J.M., Wu, X.J., Liu, W., 2015. Hominin teeth from the early Late Pleistocene site of Xujiayao, northern China. *American Journal of Physical Anthropology* 156, 224–240.

Reviewer #2 (Remarks to the Author):

Genetic analyses of contemporary humans, showing high amounts of Denisovan gene flow into Melanesian and some SE Asian populations have for a long time indicated that the Denisovans were likely present in SE Asia as well. Up until now the lack of Middle to Late Pleistocene hominin fossil remains from the region has strongly limited our ability to test this hypothesis.

In this paper, Demeter and colleagues describe the discovery of an isolated hominin lower molar from Tam Ngu Hao 2 in Laos. Using U-series, ESR and luminescence dating in a Bayesian model they establish that the specimen dates to the end of the Middle Pleistocene. They then use paleoproteomics, morphometrics and the morphology of the specimen to show that it is similar to Neanderthals and the molars of the likely Denisovan mandible from Xiahe, Tibet (comparisons with the material from Denisova cave are not easily possible, as no lower molars are known from there).

This is an important discovery, despite the rather fragmentary fossil, as it is the first specimen that could represent the hypothesized Denisovan population in the region.

In general, I feel that the paper does an admirable job at extracting information from this specimen, and I recommend its acceptance.

Reply: We thank the Reviewer for their positive comments and their appreciation of the importance of our study.

Some more detailed comments:

Line 82-95: These sentences are a bit conflicting - the authors first list a number of MP specimens from E Asia, then they say that “they were suggested to belong to an Asian sister taxon of Neanderthals, the Denisovans”, and finally they write that “The small number of fossils currently attributed to this group” listing the specimens from Denisova and Xiahe. I assume the authors mean that only the material from Denisova and Xiahe can be securely

attributed to Denisovans, while the other material is only potentially Denisovan, but I think they need to make this clearer. The way it is phrased at the moment is somewhat confusing.

Reply: Thank you for noting this. We have rephrased this part to make it clearer: Due to their combination of features, including Neanderthal-like traits, they were suggested to belong to an Asian sister taxon of Neanderthals, the Denisovans, even if their attribution to the latter group remain debated 5,11,12. The small number of fossils currently securely attributed to this group (Denisova 2, a lower left molar; Denisova 3, a distal manual phalanx; Denisova 4, an upper left M3; Denisova 8, an upper molar; and the Xiahe mandible) prohibits a clear morphological picture of the overall Denisovan morphology.

Line 96: Australian aborigines should also be mentioned here, as they have comparable amount of Denisovan introgressed DNA as Papuans

Reply: Thank you, we have added this suggestion in the text.

Line 121: “and isotopic analyses” - what kind of isotopic analyses were performed? They do not seem to be reported in the paper.

Reply: We also sampled for future isotopic analysis as investigations using geochemical proxies, such stable isotope analyses of carbon and oxygen (e.g., Farquhar et al., 1989; Cerling et al., 2015) and the novel stable zinc isotopes analyses (e.g., Jaouen et al., 2016; Bourgon et al. 2020, 2021), could elucidate this individual's overall dietary reliance and trophic level, thus shedding light into its utilization and adaptation to tropical rainforest environments.

Bourgon, N. et al. Zinc isotopes in Late Pleistocene fossil teeth from a Southeast Asian cave setting preserve paleodietary information. Proc. Natl. Acad. Sci. USA 117, 4675–4681 (2020).

Bourgon, N. et al. Trophic ecology of a Late Pleistocene early modern human from tropical Southeast Asia inferred from zinc isotopes. J. Hum. Evol., 161, p.103075 (2021).

Cerling, T. E. et al. Dietary changes of large herbivores in the Turkana Basin, Kenya from 4 to 1 Ma. Proc. Natl. Acad. Sci. USA 112, 11467–11472 (2015).

Farquhar, G. D., Ehleringer, J. R. & Hubick, K. T. Carbon isotope discrimination and photosynthesis. Annu. Rev. Plant Biol. 40, 503–537 (1989).

Line 194-197: “This is in line with previous research, which indicated that closely related hominin populations can be distinguished based on dentine and bone proteomes, while enamel proteomes are less informative in the context of close phylogenetic proximity.” If dentine proteomes would be more informative, why were they not performed? Or were these unsuccessful?

Reply: Previous research indicates that proteins survive in dental enamel beyond proteins present in dentine. Given the preservation conditions and age of the specimen, we reasoned that it is highly likely that protein degradation in the specimen is very advanced. As dental enamel proteins in principle can also shed light on hominin population relationships, we decided to analyze this tissue here.

Line 231: “and the upper molar of Denisova 4 (3D RET: 15.27)” Where does this data come from? This specimen is not listed in the SI, and there is no citation for this information either

Reply: Thanks for noticing it, we have now added the source (B. Viola, pers. comm.).

Line 241 - 255, SI Line 690 and following: The authors used a between group Principal Component Analysis to compare shape variation in the EDJ of the various groups. As pointed out by Bookstein (2019, Evolutionary Biology) and by Cardini et al (2019, cited in the paper) between group PCAs present several pitfalls when used in high p/n settings, ie. situations with a high number of variables relative to the number of specimens included. One of the largest problems is that this method can introduce spurious intergroup differences, especially in cases where the sample sizes per group differ strongly, or are low (<10). Both the GMM analyses, the one based on semi landmarks (90 semi landmarks x 3 dimensions - 270 variables), but especially the DSM approach are problematic in this context. Cardini and Polly (2020, cited in the paper) do offer some workarounds (such as cross-validated bgPCA scores), but I am not sure whether these are sufficient, especially for the DSM approach.

I want to emphasize that I do not think that this invalidates the authors' conclusions, but I would caution against overinterpreting these plots. In most of the regular PCAs (Fig S7 a,b &d) TNH2-1 plots closest to Homo erectus, which is quite interesting.

Reply: The use of between-group principal component analysis (bgPCA) was cautioned in some cases as noted by the Reviewer. However, as also indicated by the Reviewer, Cardini and Polly (2020) proposed some solutions to check if the bgPCA are reliable or if some spurious grouping is observed. As illustrated in the Fig. S7F, the normal bgPCA plot of the DSM-based approach is nearly identical to that of the cross-validated bgPCA scores shown in Fig. 3A. In addition, the landmark-based bgPCA plot in Fig. S7E is also very similar to that of the cross-validated bgPCA plot in Fig. S7G. Both DSM and landmark analyses give comparable results, even if DSM analyses better discriminate between groups. In addition, we conducted canonical variate analyses (CVA) based on a limited number of PC scores (18 and 24 for the landmark and DSM analyses respectively, which is much lower than the number of specimens exceeding 150 teeth). The CVA (Fig. 3 and Fig. S7J) and the cross-validated CVA (Fig. S7K-L) also show that group separation is not spurious and separate the three groups (even more than in the bgPCA). All these analyses, based on two kinds of data (landmark Procrustes residuals and momenta of DSM analyses), and on two kinds of discriminant methods (bgPCA and CVA), and using cross-validation, all show similar results: the three comparative groups are relatively well discriminated and the TNH2-1 and Xiahe specimens fall in between H. erectus and Neanderthals.

Regarding the position of TNH2-1 in the analyses, and in particular in the PCA, the observation of the Reviewer is only partially true. It is true that along PC1 that represents around 30% of

the total variance in both landmarks- and DSM-based analyses. However, this is mostly due to the fact that TNH2-1 and the two Xiahe molars are lower than in European Neanderthals and fossil and recent modern humans. However, along PC2 (representing 10 to 15% of the total variance), TNH2-1 (and to a lesser extent the Xiahe molars) falls outside the variation of H. erectus and shows a wider occlusal basin and higher dentine horns than in the latter taxon. The difficulty to interpret PCA plots lies in the fact that many of the first components (up to PC4 or 5, or even more) include relevant shape information. The use of bgPCA and CVA enable concentrating all shape information (or most of it) into a bi-dimensional plot directly interpretable, and more importantly, to conduct statistical analyses and compute posterior probabilities for the specimens from Cobra Cave and Baishiya Karst Cave. All in all, these specimens show a Neanderthal-like EDJ but with a lower topography.

Line 246: “Along CN2” Should this be along CV2?

Reply: Thanks for noticing this mistake, we corrected it.

Line 252: “along CV’ ” Should this be CV1?

Reply: Thank you, we have corrected it.

Line 273-275: “The similarities with Neanderthals that we observe do not preclude TNH2-1 from belonging to this taxon and would make it the south-eastern-most Neanderthal fossil ever discovered.” I find this sentence confusing. Why would similarities preclude a specimen from belonging to a taxon? Do the authors mean differences?

Reply: True, we replaced similarities by differences.

Supplementary materials

SI Line 609-11: That the metameric variation in Denisovans is different from other groups is also supported by the difficulty in identification of the Denisova 4 and 8 molars as M2s or M3s.

Reply: Absolutely, thanks. We have added this comment in the Supplementary Materials file.

SI Fig S5. : “Denisovan M1”, “Denisovan M2”: It might be better to label these as Xiahe 1, M1, and Xiahe 1, M2.

Reply: We have modified the label of these teeth.

SI Fig S6: M1s and M2s are differentiated by the fill colour in these box and whisker plots. This does not work for the AMPH group, which is represented by a single specimen (I assume this

is the M2 of Xiahe) and thus has no box. It would be the easiest to simply label this as “Xiahe M2”.

Reply: We have modified the label.

SI Fig S9., A-E: There seems to be some error with this image, panels A-E are only visible as a thin strip along the upper part of the page. I assume this happened after submission (and is not the fault of the authors), but was the case every time I downloaded the pdf.

Reply: A new picture has been uploaded for better clarity.

SI Line 970: “Halong cave” Should be Hualong cave

Reply: Thank you, we corrected this typo.

Reviewer #3 (Remarks to the Author):

The paper deals with the discovery of a human tooth found in a Middle Pleistocene breccia block in a cave site of Laos.

I will review only the dating part after some general comments on the paper.

First I would like to mention that no prehistoric context is given in the main text. The interest of the paper would be largely increased by presenting the “pre-hoabinian” lithic artefacts for instance.

Reply: We thank the Reviewer for this suggestion, however, considering the very old age of the tooth 164-131 kyr, presenting what is the “pre-Hoabinhian” culture might not be relevant in this case as a too young culture. The oldest pre-Hoabinhians remains date around 43.5 kyr and are found at the Xiaodong rock-shelter in China. Unfortunately, no other pre-Hoabinhian remains contemporaneous of our tooth has ever been found in the region. Much older lithic industry is found in China and is characterized by bifaces that are totally different from the Hoabinhian culture. For all of these reasons, we respectfully think that it wouldn't be relevant here to present in the main text the pre-Hoabinhian industries. However, we do mention in the Supplementary Material under “Geology” about the presence of some Hoabinhian industries found in another cave just nearby:

“The main entrance overlooks the road and the alluvial plain at a height of 34 m. Another cave (Tam Ngu Hao 1, Balcony Cave) is located ~10 m below the main entrance to Cobra Cave and has yielded numerous Hoabinhian stone tools as yet unpublished though similar to those identified in the nearby Tam Hang rock shelter^{1,2}. The lithic assemblage at Balcony Cave is mixed with fragments of freshwater shells and some terrestrial molluscs on two occupation levels, the oldest of which dates to 5000 years BP.”

I remain skeptical with the stratigraphic sketch presented in the paper. I am not sure that we are in presence of two formations due to different hydrogeologic phases. I would be inclined for a karstic underdrawing of a single formation of which only the finest particules have filled the lower part, the coarsest part being more easily carbonated.

Reply: We thank the Reviewer for their comments on the stratigraphy of the cave and the formation of the sediments. The sediments from LU1 filling the lower half of the chamber are strikingly different from those of the carbonate-bonded breccia (LU2) above, with a subtle yet clear interface separating them. On the basis of our systematic lithological analyses we have recorded these as two separate layers laid down under very different environments of deposition (primarily related to depositional energy and environment). Given the cramped, awkward and dark space in this chamber it was impossible to capture a satisfactory image of the erosional interface present between these units, but we have observed and recorded this in the exposed profile in an area where the two layers are still intact and form a direct relationship with one another and have annotated this in a clearer way on Figure 1b (dotted red line).

All of our detailed macro- and micro-analyses strongly indicate that the fine-grained particles of the basal layer are consistent with deposition under low-energy hydrological conditions, while the coarse and poorly sorted breccia was laid down in a higher energy environment that dumped material en masse and eroded the upper part of the basal layer, and so we do not believe that they can be grouped together as a single formation. However, we do agree with the Reviewer that the upper layer has become preferentially carbonate-cemented, most likely due to the availability of calcite in this material and post-depositional conditions.

The layers are poorly described and it seems that taphonomic considerations are not really taken into account.

Reply: We would like to point the Reviewer to our detailed descriptions of both the macro- and micro-stratigraphy of the two primary layers (and overlying carbonates) in the supplementary materials. Here we describe in detail the geomorphological context of the site and the cave morphometry, the stratigraphic architecture and detailed sedimentological characteristics and associated environments of deposition, as well as the microstratigraphy of the sediments and inferred taphonomic history of the fossils. We feel that this attention to detail, down to the micro-level, regarding the formation of the site and the sediments, and the taphonomic history of the fossils, far exceeds the usual standards for a paper reporting hominin fossils such as this. Indeed, we undertook these additional analyses for the avoidance of doubt as to the taphonomic history and context of the fossils, and our data suggest that these fossils form what is largely a single detrital assemblage of material deposited as inclusions in a mass debris flow event, as stated in the main paper and supplementary materials.

In relation to the taphonomic consideration, we added in the new version some precisions about preservation of faunal remains. "Their analyses reveal typical taphonomic pathways of assemblages from karstic systems in terms of representation of specimens and type of damage. Due to the high depositional energy (LU2) in the cave, only teeth of large mammals are present in the assemblage, and we note the absence of small and light teeth of any

microvertebrates. Moreover, most teeth are gnawed by porcupines, known to be a major accumulating agent in the region 21. Therefore, the poor preservation of specimens as shown in Fig. S15, precludes identification to the species level for most of the recorded taxa.

Fauna is associated to those of East Asia. The most recent stegodon is dated in China to less than 20 ka should at least be mentioned (reference under).

A. Ma, H. Tang, On the discovery and significance of a Holocene Ailuropoda–Stegodon fauna from Jinhua, Zhejinag, *Vertebr. Palasiat.* 30 (1992) 295–312.

Reply: Thanks to the Reviewer for sharing this work of Ma and Tang (1992). These authors suggested that Stegodon orientalis persisted until the Neolithic period ~7800 years at the Shuanglong cave in East China, which is highly debated. Indeed, the re-examination of Turvey et al. (2013) of radiometric data of Chinese megafaunal extinction supports no evidence of Holocene survival of Stegodon orientalis (along with Megatapirus augustus, and Sus xiaozhu) at this site. The dating of ~7800 years has been obtained from an indirect C14 dating on a bovid (Bubalus sp.) bone sample, a taxon present during the Holocene but that remains to be confirmed if this dated bovid sample was associated with the archaic megafauna at the Shuanglong site.

S.T. Turvey, H. Tong, A.J. Stuart, A.M. Lister. *Holocene survival of Late Pleistocene megafauna in China: a critical review of the evidence. Quat. Sci. Reviews* 76, 156-166 (2013).

In addition to this point, we also added in the main Text some precisions about environment:

The archaic Stegodon persisted in Asia most likely until the end of the Late Pleistocene (Turvey et al., 2016). Some herbivores of the fauna, Tapirus, Stegodon, Rhinocerotidae, were animals adapted to canopied woodlands in the area, whereas some others, Bos, small-sized Caprinae, large-sized Cervidae (possibly Rusa unicolor) are all known to show a great variability in their feeding behavior from closed and intermediate forests to open grassland (Bacon et al., 2021).

A.M. Bacon, N. Bourgon, F. Welker, E. Cappellini, D. Fiorillo, O. Tombret, TMH. Nguyen., AT. Nguyen, T. Sayavonkhamdy, V. Souksavatdy, P. Sichanthongtip, P.O. Antoine, P. Durringer, J.L. Ponche, K. Westaway, R. Joannes Boyau, B. Quentin, E. Suzzoni, S. Frangeul, E. Patole Edoumba, A. Zachwieja, L. Shackelford, F. Demeter, J.J. Hublin, E. Dufour. *A multi proxy approach to exploring Homo sapiens' arrival, environments and adaptations in Southeast Asia. Scientific Reports* 11, 21080 (2021).

Did the authors measure Radon (222Rn) by Gamma ray spectrometry ? You mentioned 210Pb. Did you measure it at 46 Kev? If no, the best possibility would be to measure the radon by using 214Pb at 352kev of 214Bi at 609 kev.

Reply: ^{222}Rn was measured indirectly by comparing the ^{210}Pb deficiency when compared to ^{226}Ra . This indicated that LCC2 underwent a 5.45% greater deficiency than LCC1 (22% deficiency compared to 16.55% deficiency respectively) that could be attributed to radon loss. Similarly, in the thick source alpha counting the comparison of unsealed counts to sealed counts revealed a larger increase in count rate for LCC2 (0.21 cnts/ks) compared to LCC1 (0.041 cnts/ks) indicating slightly more radon loss. So we agree that there has been greater radon loss in the LCC2 unit probably due to increased clast size compared to LCC1. However, LCC2 has the larger dose rate and increased radon loss implies that the dose rate should decrease so this cannot be the main reason for the difference in dosimetry between the samples. Instead, the largest difference in dosimetry can be attributed to the large increase in ^{40}K (increases from 467 to 889 Bg/Kg), which we have attributed to the presence of weathered granitic clasts as stayed in the discussion.

I do not see any relationship between radium loss and increase of clasts! Could you be more explicit please? According to what I measured, Even in the closest systems like some uranium standard, I observed at least 17% radon loss (I talk about radon). The explanation on the difference dose due to K seems more reasonable.

Reply: We have added the point about radon loss and clast size into the discussion of the difference in dose rates but we still argue that the main difference is the large increase in ^{40}K .

Does the important overdispersion values may suggest a pluri-origin of your minerals? Fig S10 is of very bad quality and it is not possible to read what is written for G,H, I schemes. Thank you to improve for clarity.

Reply: The overdispersion for LCC1 at 39% is just outside that of a normal distribution of well bleached grains but it is not high enough to indicate that the feldspars are from multiple sedimentary origins. More likely it indicates that the grains received a slight partial bleaching during deposition. The fact that the younger LCC2 sample has a lower overdispersion at 26% indicates that the amount of partial bleaching decreased (ie the grains received a slightly better bleaching exposure) towards the end of the depositional event. In comparison the oldest unit received the highest amount of bleaching exposure with 23% overdispersion, which lies within the range of a well bleached sample. However the MAM has been conservatively applied to all samples for consistency.

The quality of S10 fig has been improved for greater clarity.

The authors write “A better estimate of the most reliable correction is based on the assumption that the real detrital $^{230}\text{Th}/^{232}\text{Th}$ activity ratio should be lower than the bulk-Earth value of 0.825”. However, recent analyses in speleothems revealed higher $^{230}\text{Th}/^{232}\text{Th}$ than bulk earth (Carolin et al., 2016; Labonne et al., 2002). So what are the arguments for using such a low ratio in the paper? It would worth to add the fitting of the line described by the $^{234}\text{U}/^{232}\text{Th}$ against $^{230}\text{Th}/^{232}\text{Th}$. Frankly speaking, is it possible to get reliable corrected

ages with such high detrital components? In other terms, do the corrected ages have a real meaning in this case?

Carolin, S.A., Cobb, K.M, Lynch-Stieglitz, J., Moerman, J.W., Partin, J.W., Lejau, S., Malang, J., Clark, B., Tuen, A.A., Adkins, J.F. 2016. Northern Borneo stalagmite records reveal West Pacific hydroclimate across MIS 5 and 6. *Earth Planet. Sci. Lett.* 439: 182-193.

Labonne, M., Hillaire-Marcel, C., Ghaleb, B., Goy, J.-538 L. 2002. Multi-isotopic age of dirty speleothem calcite: an example from Altamira Cave, Spain. *Quaternary Science Reviews.* 21:1099-1110.

Reply: Unfortunately, flowstones adjacent to cave entrance are often susceptible to heavy contamination of clayish sediments during flowstone formation, resulting in the incorporation of high proportions of a detrital or non-radiogenic ^{230}Th component that needs to be extracted from the total measured ^{230}Th in order to obtain the true U-Th age of the flowstone sample. As the non-radiogenic ^{230}Th component cannot be physically separated from the radiogenic ^{230}Th component, an assumption of the bulk-Earth $^{230}\text{Th}/^{232}\text{Th}$ activity ratio of ~ 0.825 as the initial ratio was typically used by the U-series geochronologists to calculate the corrected U-Th ages. However, the use of this assumed value obviously results in an over-correction of the U-Th ages of our samples, as reflected by the fact that one sample (LCC-1B) returned a negative corrected age, i.e. an age of formation in the future, which cannot be true. Thus, in our Supplementary text, we clearly explained that we used the isochron method to define the non-radiogenic $^{230}\text{Th}/^{232}\text{Th}$ activity ratio, which is ~ 0.5 . This is a more realistic approach, which is in essence analogous to the stratigraphically-constrained approach proposed by Hellstrom (2006).

Regarding the Reviewer's comment that both Carolin et al. (2016) and Labonne et al. (2002) both reported higher than Bulk-Earth non-radiogenic $^{230}\text{Th}/^{232}\text{Th}$ activity ratios for their speleothems (those speleothems are a lot purer than ours), we wish to point out such a large variability is a very common phenomenon. Even in Hellstrom (2006), he reported that one speleothem with initial $^{230}\text{Th}/^{232}\text{Th} = 0.937 \pm 0.23$, higher than Bulk-Earth value of 0.825, whereas the other with initial $^{230}\text{Th}/^{232}\text{Th} = 0.24 \pm 0.06$, even twice lower than ours.

In fact, in our study of sediment-contaminated modern corals from inshore GBR settings (see Clark et al., 2004, Quat Geochron), we have made a detailed investigation of the origins of initial ^{230}Th in carbonates and found that initial non-radiogenic ^{230}Th is made of two major components: 1) an insoluble Th component adsorbed to terrestrially-derived sediments or particulates that were incorporated into the crystal lattices or cracks during speleothem formation or coral growth; and 2) a soluble or hydrogenous Th component dissolved in the cave seepage water feeding the speleothem growth or in the water column that was incorporated into the coral skeleton during growth. The insoluble Th component typically has low $^{230}\text{Th}/^{232}\text{Th}$ ratios (typically lower than the Bulk-Earth value, e.g. about 0.6 in the GBR region), whereas the soluble or hydrogenous Th component often has high and variable $^{230}\text{Th}/^{232}\text{Th}$ ratios (typically high than the Bulk-Earth value). In our dirty flowstones, the proportion of the insoluble Th component originated from the clayish sediments is predominant, so it is not unexpected for our samples to have a sample-specific $^{230}\text{Th}/^{232}\text{Th}$ ratio as defined by the isochron method. It is equally true that the much purer speleothems of

Carolin et al. (2016) have much higher initial $^{230}\text{Th}/^{232}\text{Th}$ ratios, as the soluble or hydrogenous Th component dissolved in the cave seepage water is dominant in their case.

The three analyzed teeth come from the same location in LU2. I suggest to recalculate the age without taking into account of any water in the enamel.

Reply: Ages were recalculated without taking into account any water in the enamel as requested by the Reviewer. Results for CC10, CC11 and CC12 show no significant differences with the calculated ages (164ka \pm 24, 149ka \pm 22 and 140ka \pm 23 respectively) compare to results with no water in enamel 163ka \pm 24, 149ka \pm 23 and 140ka \pm 22. Similarly marginal variations (within errors) can be observed for the modelled p-values for each tooth. Enamel is known to have trapped molecule of water usually in the order of 2 to3 %, hence our assumed values.

Secondly, the external dose rate ranges between 1100 and 1200 Gy/a and represents 80 to 93% of the total dose. I wonder how the empty space is taken into account in the external dose determination? In other terms what is the explanation of such an empty space in the sequence?

Reply: The fauna fossil teeth were located at a 20cm depth inside a dense breccia, nonetheless, the stratigraphy indicates that a large part of the structure had eroded away recently. Therefore, for the calculation of the external dose rate, we have modelled the parameters assuming a full 30cm sphere of surrounding sediment. If we were to model the external dose rate using the current burial configuration, the external dose rate estimation would be 699 \pm 121mGy/kyr (about ~22% less). Age estimates would be shifted to 190 \pm 28kyr, 177 \pm 26kyr and 168 \pm 27kyr for CC10, CC11 and CC12 respectively. A paragraph explaining our assumption has been added, as well as the results using a 20cm depth configuration.

Thank you to mention what % of water was used for the sediment?

Reply: The sediment water content, as well all other water contents (dentine and enamel) can already be found in table S2, that summarises all parameters and results for the US-ESR dating estimations.

Does it worth to include the CC10 age in the weighted meaning age? I do not think so! In that case we would better be in a MIS5 infilling!

Reply: It is our understanding that all ages should be included in the model, rather than selecting and/or excluding certain ages. Minimum and maximum ages are also included as such in the model and allows to have a proper representation of all the dating results obtained. We feel that excluding some ages would introduce an operator bias into the modelling.

What is the meaning of Bayesian model without a real stratigraphic sequence? Why did you suppress the LCC3 age?

Reply: We disagree that there is no stratigraphic sequence present. The basal unit at 248 +/- 31 ka is clearly older than the breccia deposit 142-136 ka, and the overlying flowstone 104 +/- 27 ka was clearly precipitated after breccia deposition. This sequence of deposition is stratigraphically correct with no time inversions. As we have employed multiple techniques to constrain the breccia and its associated fossil material the use of the Bayesian model allows the probability distributions of individual ages to be incorporated, along with the constraints imposed by stratigraphic relationships and the reported minimum or maximum nature of some of the individual age estimates. Thus, we are able to estimate an age for the tooth incorporating all age estimates rather than relying on the results of just one technique. The model estimates the age of the boundary between the basal unit to the breccia, and the breccia to the overlying flowstone. This provides a much better estimation of the age range of each unit rather relying on one age from inside the unit itself.

The LLC3 age at 248 +/- 31 ka stands alone as an age estimate for the basal unit and has been entered into the model as such. The younger age of 164 +/- 17 ka (seen in the model directly above the basal unit) is the modelled age for the boundary between the basal unit and the breccia - taking into account the uncertainty on the age estimate for the basal unit and the multiple age estimates for the breccia. Thus, the LCC3 age estimate has not been suppressed but rather we have used this age to estimate the boundary with the breccia.

REVIEWER COMMENTS

Reviewer #1 (Remarks to the Author):

From looking over the substantially revised manuscript, the authors have done an admirable job on really strengthening the study. It will make a nice addition to the literature. It should be published As Is. I look forward to not only seeing it in print, but to citing it in my own work as well. Excellent job!!

Reviewer #2 (Remarks to the Author):

The authors have addressed my previous concerns. The manuscript is excellent, and I am looking forward to reading it in print.

I noticed a typo in Figure S1, where the tooth is described as "Denisovian", it would be better to use "Denisovan" (so without the second i) here.

Reviewer #3 (Remarks to the Author):

The authors generally answered to my issues. Before accepting the paper for publication I still have some questions or comments:

The reply about what I called "Pre-Hoabinian" does not correspond to Xiaodong which is the oldest Hoabinian technocomplex, and no pre-Hoabinian, described by Ji and al., 2016. I wanted to say that the 100-200 ka period in this area could be assimilated to a "transitional period" with very few data. In this way the discovery of a human remain belonging to this period is very important and this should be contextualized either in the case of the site or at the regional scale. I understand that no lithic artefact was found in the breccia.

About the stratigraphy, the less that can be said is that the erosional interface is very subtle in the narrowest part of the infilling without visible changing of structure! Anyway I accept the explanations of the authors concerning the difference between the two units (figure 1).

About the increase in 40K, I would like to know if you observed in the sediment the presence of some manganese oxydes like cryptomelane which has a strong capacity to trap potassium. Because I do not see any reason able to explain why weathered granitic clasts would be absent in the lowest part while present in the upper one excepting maybe the size of the clasts?

I do not still understand why some feldspars grains would be more bleached than others. How do you explain the better bleaching conditions for the LCC2 sample? Do you mean that there is a possibility to get an age overestimation on the less bleached sample? Why do not use the correction method published by Lamothe et al, 2020 to circumvent anomalous fading?

References

Ji X., Kuman K., Clarke R.J., Forestier H., Li Y., Ma J., Qiu K ;, Li H., Wu Y. (2016). Quaternary International, 400, 166-174.

Lamothe M., Forget L., Brisson, F., Hardy F. (2020). Quaternary Geochronology, 57, 101062.

REPLY TO THE REVIEWERS COMMENTS

Dear reviewers, we thank you for your comments. We addressed one by one all the questions raised by Reviewer 3 below.

Reviewer #1 (Remarks to the Author):

From looking over the substantially revised manuscript, the authors have done an admirable job on really strengthening the study. It will make a nice addition to the literature. It should be published As Is. I look forward to not only seeing it in print, but to citing it in my own work as well. Excellent job!!

Reply: We thank you for your comment and congratulations.

Reviewer #2 (Remarks to the Author):

The authors have addressed my previous concerns. The manuscript is excellent, and I am looking forward to reading it in print.

I noticed a typo in Figure S1, where the tooth is described as "Denisovian", it would be better to use "Denisovan" (so without the second i) here.

Reply: We thank you for your comment and congratulations. We corrected the typo in Figure S1.

Reviewer #3 (Remarks to the Author):

The authors generally answered to my issues. Before accepting the paper for publication I still have some questions or comments:

The reply about what I called "Pre-Hoabinian" does not correspond to Xiaodong which is the oldest Hoabinian technocomplex, and no pre-Hoabinian, described by Ji and al., 2016. I wanted to say that the 100-200 ka period in this area could be assimilated to a "transitional period" with very few data. In this way the discovery of a human remain belonging to this period is very important and this should be contextualized either in the case of the site or at the regional scale. I understand that no lithic artefact was found in the breccia.

Reply: We thank you for your comment and agree with your consideration regarding the Xiaodong lithic material being the oldest Hoabinian technocomplex and not being a "Pre-Hoabinian complex. We also agree with you when you say that the 100-200 ka period could be assimilated to a "transitional period" in the region. However, as we already mentioned in our previous response, it is correct that we did not find any artefact associated with the

Denisovan tooth or with the faunal remains in the breccia. We agree that finding artefacts of that age would have been a breakthrough for both local and regional archaeology.

About the stratigraphy, the less that can be said is that the erosional interface is very subtle in the narrowest part of the infilling without visible changing of structure! Anyway I accept the explanations of the authors concerning the difference between the two units (figure 1).

Reply: We thank you for your comment.

About the increase in 40K, I would like to know if you observed in the sediment the presence of some manganese oxydes like cryptomelane which has a strong capacity to trap potassium. Because I do not see any reason able to explain why weathered granitic clasts would be absent in the lowest part while present in the upper one excepting maybe the size of the clasts?

Reply: We didn't observe manganese oxide during luminescence processing and under light optical examination, nor did we observe it at the site or in thin section. The reviewer is correct in suggesting that the weathered granite clasts were only observed in the upper clast-supported part of LU2 (clast:matrix ratio 80%). This increase in clasts relates to an increase in depositional energy associated with a flood event that deposited a coarsening upwards sedimentary profile (Supp; Geology). This implies the larger clasts were not transported during the lower-energy flows associated with the basal part of LU2 (LCC1 sample location with a clast:matrix ratio of 50%), thus it has a correspondingly lower 40K value.

I do not still understand why some feldspars grains would be more bleached than others. How do you explain the better bleaching conditions for the LCC2 sample?

Reply: The deeper LU2 sediments represent a slower flow so possibly a supersaturated debris flow that contains less bleached grains due to its colluvial nature, hence the higher (39%) OD values. In contrast, the larger more rounded clasts in the upper sections of LU2 indicates more of a fluvial discharge that presents more opportunities for bleaching of the matrix, hence the lower (26%) OD values.

Do you mean that there is a possibility to get an age overestimation on the less bleached sample?

Reply: We estimate that breccia deposition was fairly rapid, most likely deposited during a single event over the course of days or hours. Thus, the depositional age of LCC1 and LCC2 samples are coeval within errors. The fact that the ages agree within errors indicates that the difference in partial bleaching is only minor, and even if LCC1 grains are overestimating this value cannot be large otherwise the age estimates would not agree. As discussed in the OSL discussion, when taken in the context of the relationship between the sediments and fossils, it is clear that the deposition of the sediments is the last event to occur resulting in the sediment

ages being the youngest - even if they are maximum ages. The age of the sediments does not impact on the age of the fossils, which are all older.

Why do not use the correction method published by Lamothe et al, 2020 to circumvent anomalous fading?

Reply: We used the anomalous fading correction of Lamothe et al., 2003 because when this analysis was conducted in 2018-2019 this paper was not yet published. More recently we applied the 2020 correction method to the samples and obtained a similar result so we decided to retain the original correction.

REVIEWERS' COMMENTS

Reviewer #3 (Remarks to the Author):

Thank you for your answers and I think that the paper can be published in its state.